# DUSP16 promotes cancer chemoresistance through regulation of mitochondria-mediated cell death

Heng Boon Low[1,2,15], Zhen Lim Wong [1,2,15], Bangyuan Wu [1,2,3,15], Li Ren Kong [4], Chin Wen Png[1,2], Yik-Lam Cho[5], Chun-Wei Li[6], Fengchun Xiao[7], Xuan Xin[8], Henry Yang[4], Jia Min Loo[9], Fiona Yi Xin Lee[10], Iain Bee Huat Tan[10], Ramanuj DasGupta [9], Han-Ming Shen[5,11], Herbert Schwarz [2,5], Nicholas R. J. Gascoigne [1,2], Boon Cher Goh [4,12,13], Xiaohong Xu[14✉] & Yongliang Zhang [1,2✉]

Drug resistance is a major obstacle to the treatment of most human tumors. In this study, we find that dual-specificity phosphatase 16 (DUSP16) regulates resistance to chemotherapy in nasopharyngeal carcinoma, colorectal cancer, gastric and breast cancer. Cancer cells expressing higher DUSP16 are intrinsically more resistant to chemotherapy-induced cell death than cells with lower DUSP16 expression. Overexpression of *DUSP16* in cancer cells leads to increased resistance to cell death upon chemotherapy treatment. In contrast, knockdown of *DUSP16* in cancer cells increases their sensitivity to treatment. Mechanistically, DUSP16 inhibits JNK and p38 activation, thereby reducing BAX accumulation in mitochondria to reduce apoptosis. Analysis of patient survival in head & neck cancer and breast cancer patient cohorts supports DUSP16 as a marker for sensitivity to chemotherapy and therapeutic outcome. This study therefore identifies DUSP16 as a prognostic marker for the efficacy of chemotherapy, and as a therapeutic target for overcoming chemoresistance in cancer.

[1] Department of Microbiology and Immunology, Yong Loo Lin School of Medicine, National University of Singapore, Singapore, Singapore. [2] Immunology Programme, the Life Science Institute, National University of Singapore, Singapore, Singapore. [3] College of Life Science, China West Normal University, Nanchong, Sichuan, China. [4] Cancer Science Institute of Singapore, National University of Singapore, Singapore, Singapore. [5] Department of Physiology, Yong Loo Lin School of Medicine, National University of Singapore, Singapore, Singapore. [6] Department of Otorhinolaryngology, The First Affiliated Hospital of Sun Yat-Sen University, Guangzhou, China. [7] Department of Pathology, The First Affiliated Hospital of Zhejiang Chinese Medical University, Hangzhou, China. [8] Department of Mathematics, National University of Singapore, Singapore, Singapore. [9] Genome Institute of Singapore, Agency of Science Technology and Research (A*Star), Singapore, Singapore. [10] Division of Medical Oncology, National Cancer Center, Singapore, Singapore. [11] Faculty of Health Sciences, University of Macau, Macau, China. [12] Department of Haematology-Oncology, National University Cancer Institute, Singapore, Singapore. [13] Department of Pharmacology, Yong Loo Lin School of Medicine, National University of Singapore, Singapore, Singapore. [14] Department of Breast Surgery, The First Affiliated Hospital of Zhejiang Chinese Medical University, Hangzhou, China. [15]These authors contributed equally: Heng Boon Low, Zhen Lim Wong, Bangyuan Wu. ✉email: miczy@nus.edu.sg; samsurgery@126.com

Cancer drug resistance is a complex phenomenon that is influenced by various mechanisms. Drug resistance may arise intrinsically from host factors or may be acquired by genetic or epigenetic alterations in the cancer cells[1]. Cancers are known to exhibit micro-clonality with a high degree of genetic heterogeneity making it possible for resistant cells to continue proliferating even in the presence of therapeutic treatments[2]. This genetic heterogeneity may confer upon cancer cells the ability to regulate rates of drug efflux, DNA damage repair processes, and signaling pathways affecting cell survival and death[3].

Drugs including platinum-based drugs and others such as 5-fluorouracil (5-FU) and epirubicin remain the major chemotherapeutic agents used in the treatment of a wide variety of solid malignancies, including colorectal, ovarian, lung, and head and neck cancer[4,5]. There are three platinum-based drugs, cisplatin, carboplatin, and oxaliplatin, which are used throughout the world, with four additional drugs, heptaplatin, lobaplatin, miriplatin, and nedaplatin, having regulatory approval in individual countries such as Korea, China, and Japan[6]. Cisplatin is currently one of the most potent chemotherapeutic drugs and is widely used either alone or in combination with other drugs in cancer treatment. For instance, it is the most frequently used therapeutic agent in treating metastatic nasopharyngeal carcinoma (NPC)[7], a malignancy particularly endemic to Southeast Asia and Southern China[8,9]. In the treatment and management of colorectal carcinoma, platinum compound-based chemotherapeutic agents, including cisplatin, are used in combination with other drugs[10,11]. Studies have also supported the use of platinum-based chemotherapy in breast cancer, showing cytostatic effects of cisplatin in clinical trials[12], and as part of neoadjuvant chemotherapy leading to remission[13]. The cytotoxicity of cisplatin and other platinum-based drugs is mediated by their interaction with DNA to form DNA adducts, leading to DNA damages that activate signal transduction pathways and consequently cell apoptosis[14]. 5-FU, on the other hand, is an apyrimidine analog that inhibits thymidylate synthase and can also misincorporate into DNA during polynucleotide biosynthesis, which leads to DNA damage and apoptosis[15]. Epirubicin, an anthracycline drug widely employed to treat solid tumors, acts through intercalating between nucleic acid base pairs to inhibit both DNA and RNA biosynthesis and also triggering DNA cleavage and promoting the production of reactive oxygen species, eventually leading to cell death[16].

Chemotherapy treatment of patients with solid tumors or cancers of soft tissue, bones, muscles, or blood vessels has led to better prognosis and increased life expectancy[17]. However, resistance to therapies, including intrinsic resistance of patients and acquired resistance, can lead to therapeutic failure and is the main limitation of these drugs in cancer treatment[18]. As such, a great deal of effort has been dedicated to elucidating the mechanisms underlying the resistance to these cytotoxic drugs. Multiple mechanisms have been found to play roles in the resistance to these therapies, including reduced intracellular accumulation and increased sequestration of the drug, increased repair/tolerance to DNA damage, and manipulation of signal transduction pathways involving drug-induced apoptosis by cancer cells[4,5]. However, knowledge on strategies to overcome such limitations and biomarkers that can predict patient response to chemotherapy is scarce.

The mitogen-activated protein kinase (MAPK) pathways, including the extracellular signal-regulated kinase (ERK), the c-Jun N-terminal kinase (JNK), and the p38 kinase, are evolutionarily conserved signaling pathways that regulate cell proliferation, survival, and apoptosis in cancer[19–23]. Previous studies have implicated the MAPK signaling pathways in responses to chemotherapeutic agents. For instance, ERK1/2 activation has been shown to promote cancer cell survival in ovarian cancer[24,25], melanoma[26], cervical cancer[27], myeloid leukemia[28], and gastric cancer[29] cells in response to cisplatin treatment. However, there is also evidence showing that ERK1/2 activation is necessary for cisplatin-mediated apoptosis of cells of cervical cancer[30,31], osteosarcoma and neuroblastoma[32], glioma[33], NPC[34], and human small cell lung cancer[35]. Activation of the p38 pathway by cisplatin has been observed in different types of cancer cells[36]. Inhibition of p38 suppressed cisplatin-induced apoptosis while prolonged p38 activation has been associated with increased sensitivity to cisplatin-induced apoptosis in cervical[37], ovarian[38], and breast cancers[39]. Similarly, activation of the JNK pathway plays a major role in mediating apoptosis by cisplatin treatment[40,41]. Interestingly, the oncogenic EBV gene, LMP-1, has been found to activate the JNK pathway and promote cisplatin-induced caspase activation and apoptosis[42–44]. Moreover, inhibition of cytokeratin 8 in NPC cell lines increased cancer cell sensitivity to cisplatin by activating the JNK pathway[45–47]. In addition, the MAPKs have been shown to mediate sensitivity of a variety of cancer cells such as colorectal cancer, pancreatic cancer, and breast cancer to other therapeutic drugs including 5-FU and epirubicin[15,48–50].

The activities of MAPKs are mainly regulated by a family of proteins called dual-specificity phosphatases (DUSPs) which are also known as MAP kinase phosphatases (MKPs)[51,52]. DUSP16/MKP7, initially identified as a protein to inhibit both p38 and JNK[53], can also inactivate ERK in immune cells including macrophages and CD4+ T cells[54–56]. Action of DUSP16 has been implicated in leukemia[57], NSCL cancer[58,59], prostate carcinoma[60], and Burkitt's lymphoma[61], possibly through inactivation of JNK[57]. Interestingly, silencing of DUSP16 in Burkitt's lymphoma cells has been shown to increase JNK activity and enhance sensitivity to doxorubicin, sorbitol, and cisplatin[61]. However, whether DUSP16 regulates cancer sensitivity to platinum-based therapies and other drugs is unclear.

In this study, we show that cancer cells with higher DUSP16 expression were more resistant to chemotherapy-mediated cell death. Overexpression of *DUSP16* in cancer cells, including NPC, CRC, breast and gastric cancer cells, resulted in reduced cell death upon treatment with chemotherapy drugs, including cisplatin, carboplatin, oxaliplatin, fluorouracil (5-FU), and epirubicin in vitro and in vivo; whereas CRISP/Cas9-mediated knockout of *DUSP16* in cancer cells led to an increased sensitivity to these drugs. The correlation between the levels of DUSP16 and the sensitivity to treatment is associated with changes in JNK/p38 activation, BAX accumulation in mitochondria, cytochrome *c* release, and the activation of caspases 9/3.

## Results

**Levels of DUSP16 in cancer cells are inversely associated with sensitivity to cisplatin.** To investigate the possible function of DUSP16 in cancer, expression of endogenous DUSP16 in four NPC cell lines, including C666-1, CNE-1, HK-1, and HONE-1, was assessed by quantitative real-time PCR (qPCR) and western blot analysis. It was found that the undifferentiated NPC cell line, C666-1, showed at least twofold higher mRNA expression of *DUSP16* than the differentiated NPC cell lines CNE-1, HK-1, and HONE-1 (Fig. 1A). Protein expression of DUSP16 was only detected in C666-1 and HK-1 cells, but not in CNE-1 and HONE-1 cells (Fig. 1B), and a higher level of protein expression was detected in C666-1 cells than in HK-1 cells.

Next, C666-1 and HK-1 cells were treated with cisplatin, a chemotherapeutic drug most often used in treatment of NPC, to examine any changes in expression of DUSP16. *DUSP16* mRNA expression could be strongly induced by cisplatin in HK-1 cells,

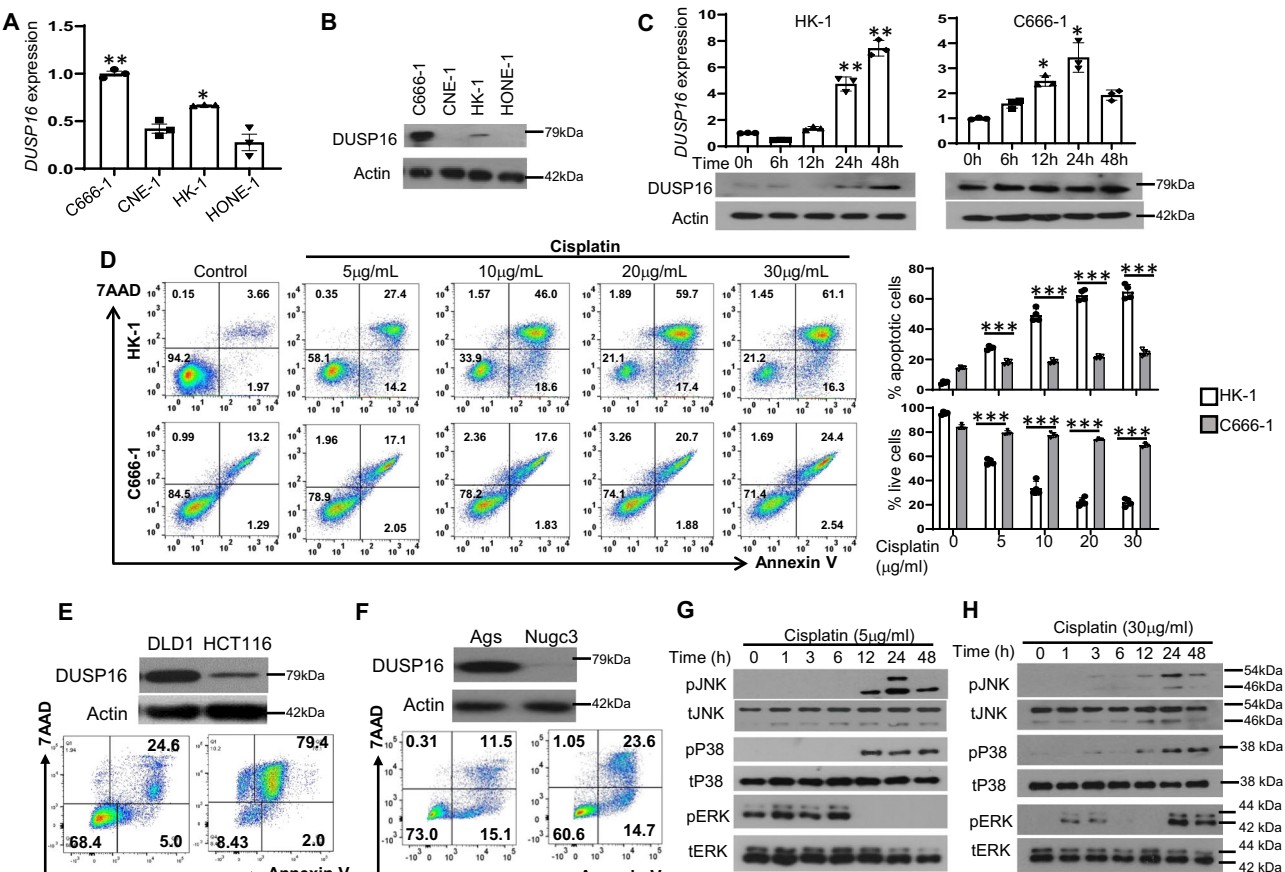

**Fig. 1 DUSP16 protein expression level is associated with cancer cell sensitivity to cisplatin. A**, **B** Basal expression levels of *DUSP16* in NPC cell lines were determined by quantitative real-time PCR (qPCR). Data were expressed as means ± standard error of the mean (mean ± SEM, *n* = 3 biologically independent samples). Two-tail unpaired *t*-test showed significantly higher *DUSP16* expression in C666-1 and HK-1 cells compared to that in HONE-1 cells. *$P < 0.05$, **$P < 0.001$ (**A**). DUSP16 protein expression in these cells was determined by western blot analysis. The data are representative of two experiments with similar results (**B**). **C** Expression of DUSP16 in HK-1 and C666-1 in response to cisplatin treatment was determined by qPCR and western blot analysis. Induction of *DUSP16* mRNA expression at each time point was expressed as means ± SEM (*n* = 3 biologically independent samples). Statistical analysis was performed using two-tailed unpaired *t*-test. *$P < 0.05$, **$P < 0.001$. **D** Flow cytometric analysis measuring apoptosis in HK-1 and C666-1 cell lines at 48 h after cisplatin treatment. Bar charts show the percentage of apoptotic (AnnexinV⁺7AAD⁺) and surviving (AnnexinV⁻7AAD⁻) cells, respectively, in both untreated and cisplatin-treated groups with mean ± SEM (*n* = 3 biologically independent samples). Statistical analysis was performed using two-tailed unpaired *t*-test. ***$P < 0.0001$. The data are representative of three experiments with similar results. **E**, **F** DUSP16 protein expression in colorectal cancer cell DLD-1 and HCT116 (**E**) and in gastric cancer cell Ags and Nugc3 was evaluated by western blot analysis. The data are representative of 3 experiments with similar results. **G**, **H** Phospho-JNK (pJNK), phospho-P38 (pP38), phospho-ERK (pERK) and their total protein expression (tJNK, tP38, and tERK) in HK-1 (**G**), and C666-1 (**H**) cells upon cisplatin treatment was examined by western blot analysis. The data are representative of 3 experiments with similar results. Source data are provided as a Source data file.

but was only weakly induced in C666-1 cells (Fig. 1C). Expression of DUSP16 protein was increased at 24 and 48 h upon cisplatin treatment in HK-1 cells, but not in C666-1 cells which constitutively express a high level of DUSP16 protein (Fig. 1C). These results suggest the possible involvement of DUSP16 in NPC response to cisplatin.

To test if the level of DUSP16 expression could influence the sensitivity of NPC cells to cisplatin, C666-1 and HK-1 cells were treated with increasing concentrations of cisplatin for 24 or 48 h to assess apoptosis by AnnexinV/7AAD staining and flow cytometric analysis. Increased percentages of apoptotic cells were observed in both HK-1 and C666-1 cells with increasing concentrations of cisplatin at both 24 and 48 h after treatment (Fig. 1D and Supplementary Fig. 1A-B). In addition, a greater increase in apoptosis was observed in HK-1 cells than in C666-1 cells in response to all concentrations of cisplatin. For instance, upon 5 μg/mL cisplatin treatment, the percentage of total apoptotic cells in HK-1 was 41.6% (including both AnnexinV

single positive and AnnexinV/7AAD double-positive cells), but was only 19% in C666-1 (Fig. 1D). Increasing cisplatin concentration from 5 to 10 μg/mL resulted in 64.6% of apoptotic cells in HK-1 cells, but only 19.4% in C666-1. Moreover, 48 h after treatment, a significant percentage of apoptosis was detected only at the concentration of 30 μg/mL in C666-1 cells compared to control cells, whereas significant apoptosis was detected in HK-1 cells at all concentrations of cisplatin compared to the control group. Therefore, C666-1 cells, which express a higher level of DUSP16, are more resistant to cisplatin treatment than HK-1 cells which express a much lower level of this molecule. This suggests that DUSP16 may be involved in cisplatin resistance of NPC. For subsequent experiments, the lowest cisplatin concentration to cause significant levels of apoptosis in each cell type was used: 5 and 30 μg/mL for HK-1 and C666-1 cells, respectively.

To test if DUSP16 regulates sensitivity to cisplatin in other cancer cells, its expression in CRC cell lines DLD-1 and HCT116 (Fig. 1E and Supplementary Fig. 1C), and the gastric cancer cell

lines Ags and Nugc3 (Fig. 1F and Supplementary Fig. 1D) was examined. Higher expression of DUSP16 was detected in DLD-1, and Ags than in HCT116, and Nugc3, respectively, which was associated with more resistance to cisplatin in DLD-1 or Ags cells compared to HCT116 or Nugc3 cells, respectively. These results suggest that levels of DUSP16 in cancer cells may inversely be correlated with sensitivity to cisplatin.

To understand the targets of DUSP16 in NPC cisplatin resistance, the activation of MAPKs, including ERK, JNK, and p38, was assessed in both HK-1 and C666-1 cells upon treatment with 5 and 30 µg/mL cisplatin, respectively. Both JNK and p38 activation were induced in HK-1 cells in response to 5 µg/mL of cisplatin treatment, with similar activation patterns (Fig. 1G). ERK was constitutively activated in HK-1 cells and its activation was increased at 1, 3, and 6 h following treatment. Treating C666-1 cells with 30 µg/mL of cisplatin induced similar patterns of JNK and p38 activation as those observed in HK-1 cells treated with 5 µg/mL of cisplatin (Fig. 1G, H); activation was observed at 12 h post treatment and reached the highest activation at 24 h. JNK activation was reduced subsequently by 48 h, whereas the activation of p38 remained the same as that at 24 h. Unlike HK-1 cells, ERK was not constitutively active in C666-1 cells, and cisplatin induced weak activation of ERK at early time points,

including at 1 and 3 h upon treatment, and stronger activation at late time points, including at 24 and 48 h (Fig. 1H).

**Overexpression of *DUSP16* in cancer cells enhances resistance to apoptosis induced by cisplatin.** To assess the potential role of DUSP16 in cancer cells, we transfected a full-length human *DUSP16* cDNA expressing plasmid into various types of cancer cells, including NPC HK-1 cells, colorectal cancer DLD-1, and HCT116 cells, gastric cancer Nugc3, and breast cancer MDA-MB-231 cells to generate *DUSP16* stably transfected clones. Increased DUSP16 expression at both mRNA and protein levels in *DUSP16* stably transfected clones compared to vector-transfected cells shows the successful generation of DUSP16-overpressing cells (Figs. 2A, D, G and Supplementary Fig. 2A).

To assess the role of DUSP16 in the regulation of cisplatin-mediated responses in HK-1 cells, Vector- or DUSP16-expressing HK-1 cells were treated with 5 µg/mL cisplatin to examine cell apoptosis. After 48 h of cisplatin treatment, around 60% of vector-transfected cells were apoptotic (including AnnexinV$^+$7AAD$^-$ and AnnexinV$^+$7AAD$^+$ cells) (Fig. 2B). In contrast, the percentage of apoptotic cells was only around 34% in *DUSP16*-expressing cells, which was significantly lower than that in vector-

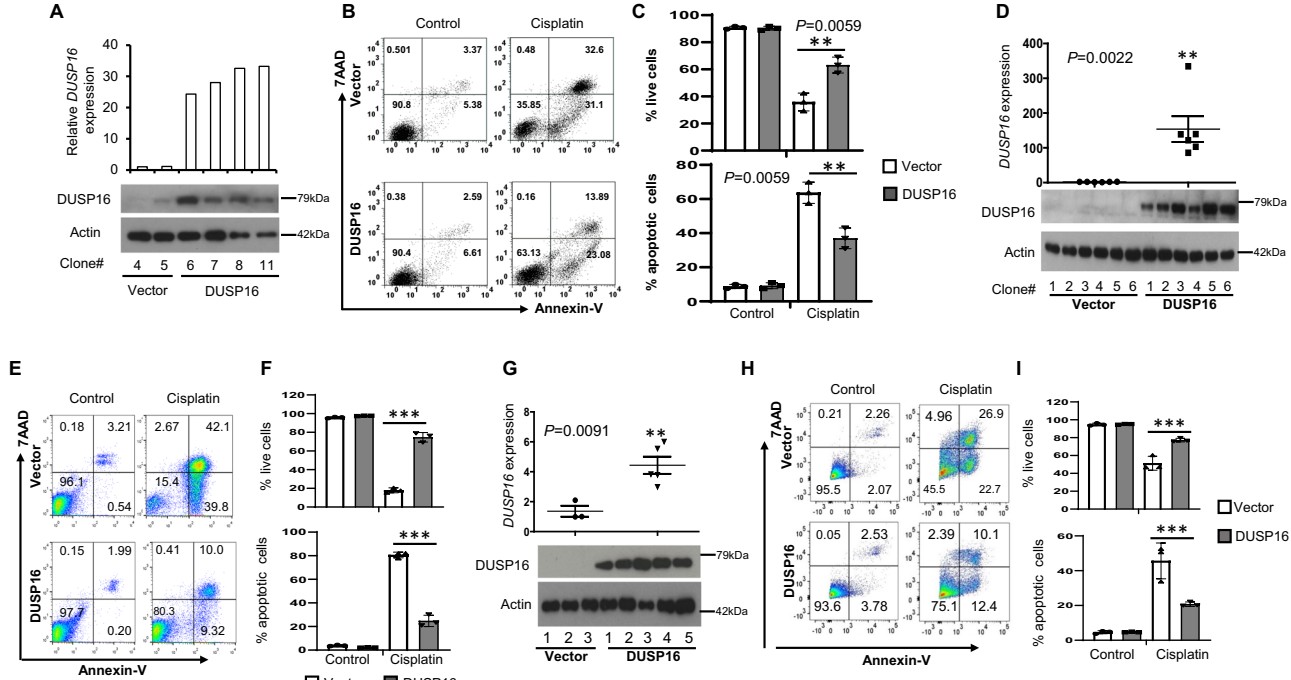

**Fig. 2 Overexpression of *DUSP16* in various types of cancer cells results in increased resistance to cisplatin. A** mRNA and protein levels of *DUSP16* in HK-1 cells transfected with vector or human *DUSP16* cDNA were quantified by qPCR and western blot analysis, respectively. Bar chart shows mRNA expression of vector (n = 2) and *DUSP16* (n = 4) expressing clones. **B, C** Comparison of apoptosis induction between vector- and *DUSP16*-transfected cells after cisplatin treatment was carried out by staining the cells with AnnexinV and 7AAD followed by flow cytometry analysis (**B**). Bar charts show the percentage of apoptotic (both AnnexinV$^+$7AAD$^+$ and AnnexinV$^+$7AAD$^-$ cells) and surviving cells, respectively, in both untreated and cisplatin-treated groups with mean ± SEM, n = 3 biologically independent samples (**C**). Statistical analysis was performed using two-tailed unpaired *t*-test. The data are representative of 3 experiments with similar results. **D** *DUSP16* mRNA expression in DLD-1 cells transfected with vector (n = 6) or human *DUSP16* cDNA (n = 6) quantified by qPCR. Statistical analysis was performed using two-tailed unpaired *t*-test. DUSP16 protein expression in these clones was assessed by Western blot. **E, F** Apoptosis induction between the vector- and *DUSP16*-transfected DLD-1 cells after cisplatin treatment was analyzed (**E**) and bar charts show the percentage of apoptotic and surviving cells in both untreated and cisplatin-treated groups with mean ± SEM, n = 3 biologically independent samples (**F**). Statistical analysis was performed using two-tailed unpaired *t*-test. ***P < 0.0001. **G** *DUSP16* mRNA and protein expression in Nugc3 cells transfected with vector or human *DUSP* cDNA was determined by qPCR and immunoblotting, respectively. Dot plot shows *DUSP16* mRNA expression in vector (n = 3) and *DUSP16* (n = 5) expressing clones. Statistical analysis was performed using two-tailed unpaired *t*-test. **H, I** Apoptosis induction between vector- and *DUSP16*-transfected Nugc3 cells after cisplatin treatment was analyzed and bar charts (**I**) show the percentage of apoptotic and surviving cells in untreated and cisplatin-treated cells, respectively, with mean ± SEM, n = 3 biologically independent samples. Statistical analysis was performed using two-tailed unpaired *t*-test. ***P < 0.0001. The data are representative of 3 experiments with similar results. Source data are provided as a Source data file.

expressing cells. Conversely, the number of surviving (AnnexinV⁻7AAD⁻) cells was approximately twofold higher in *DUSP16*-expressing cells than that of vector-expressing cells (Vector-HK-1 vs DUSP16-HK-1: 35.85% vs 63.13%, *p* < 0.01, Fig. 2B, C).

Comparison of apoptosis induction between vector- and *DUSP16*-overexpressing CRC DLD-1, gastric Nugc3 cells, and breast cancer MDA-MB-231 cells showed similar trends to that observed in HK-1 cells. About 80% apoptotic cells were detected in DLD-1 vector-transfected cells after 48 h of cisplatin (30 μg/mL) treatment, while only 19% of apoptotic cells were detected in the *DUSP16*-overexpressing DLD-1 cells (Fig. 2E, F). Conversely, the percentage of cell survival was significantly lower in the vector-transfected DLD-1 cells compared to the *DUSP16*-overexpressing DLD-1 cells (Vector-DLD-1 vs *DUSP16*-DLD-1: 15% vs 80%, *p* < 0.01, Fig. 2E, F). Similarly, vector-transfected Nugc3 cells showed higher induction of apoptosis (Vector-Nugc3 vs DUSP16-Nugc3: 49% vs 22% *p* < 0.0001, Fig. 2H, I) and lower percentage of survival (Vector-Nugc3 vs DUSP16-Nugc3: 45% vs 75% *p* < 0.0001, Fig. 2H, I) compared to the *DUSP16*-overexpressing Nugc3 cells after 48 h of cisplatin (10 μg/mL) treatment. Vector-transfected MDA-MB-231 cells also showed higher induction of apoptosis (Vector-MDA-MB-231 vs DUSP16-MDA-MB-231: 28% vs 16% *p* < 0.01, Fig S2B-C) and lower percentage of survival (Vector-MDA-MB-231 vs DUSP16-MDA-MB-231: 70% vs 84% *p* < 0.01, Supplementary Fig. 2B-C) compared to the *DUSP16*-overexpressing MDA-MB-231 cells.

**Higher levels of DUSP16 are correlated with increased resistance to various chemotherapeutic agents.** To test whether DUSP16 regulates the response of cancer cells to other chemotherapy drugs, we treated NPC cells HK-1 and C666-1 with carboplatin which is used for treating NPC[62], and CRC cells DLD-1 and HCT116 with oxaliplatin which is used in the management of metastatic colorectal cancer[63]. Compared to HK-1 cells, the DUSP16-higher C666-1 cells exhibited significantly reduced apoptosis in response to carboplatin treatment (Fig. 3A). Overexpression of *DUSP16* in HK-1 cells resulted in increased resistance to both oxaliplatin and carboplatin (Fig. 3B). Similarly, in DLD-1 cells which have higher DUSP16 expression showed reduced apoptosis upon oxaliplatin treatment compared to HCT116 cells (Fig. 3C), and overexpression of *DUSP16* in HCT116 cells resulted in a significant reduction in apoptosis in response to both oxliplatin and carboplatin treatment compared vector-expressing cells (Fig. 3D). Similar results were observed in DUSP16-overexpressing DLD-1 cells in response to cisplatin, carboplatin, and oxalipatin treatments (Supplementary Fig. 3A-B). These results demonstrated that *DUSP16* expression is positively associated with resistance to chemotherapy drugs in cancer cells.

**DUSP16 regulates JNK and p38 activation in response to chemotherapeutic drug treatment.** MAPKs are important regulators of cell apoptosis[64]. Next, the activation of the MAPKs in

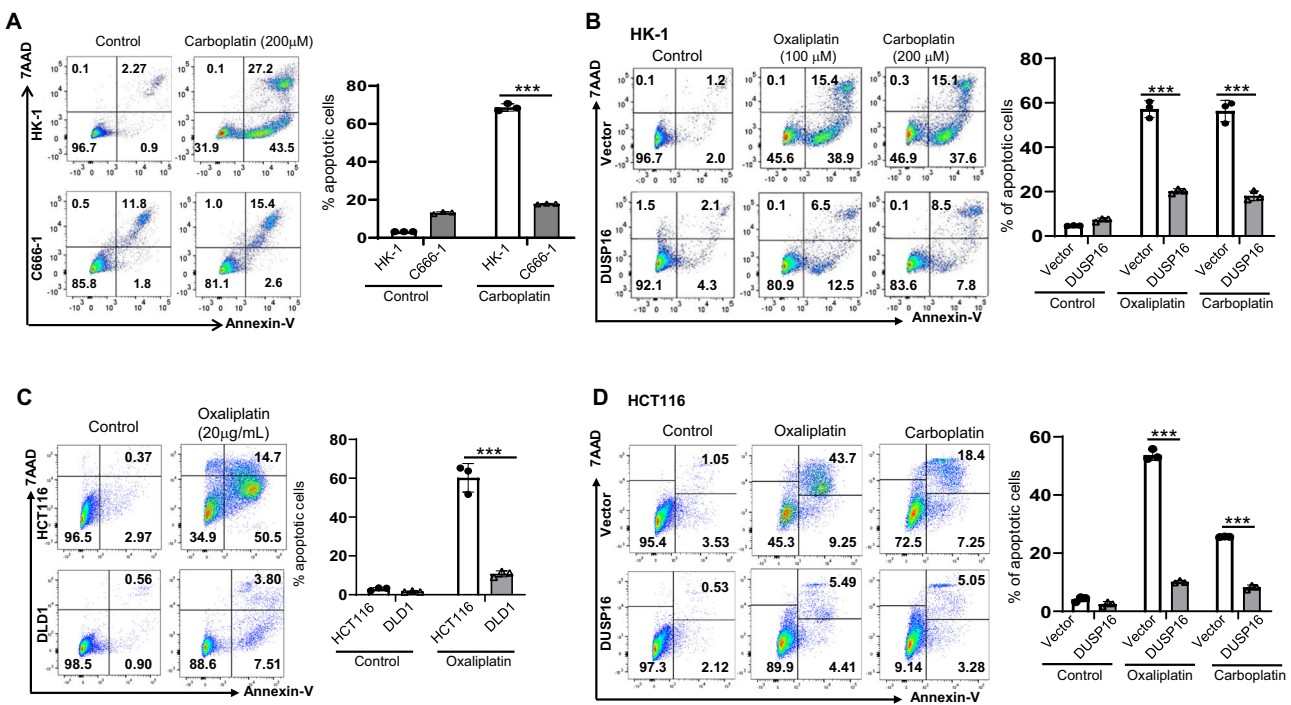

**Fig. 3 Levels of DUSP16 in various types of cancer cells are associated with sensitivity to carboplatin and oxaliplatin. A** Comparison of apoptosis induction between HK-1 and C666-1 cells in response to carboplatin treatment by flow cytometry analysis. Bar chart shows the percentage of apoptotic cells with mean values ± standard error of the mean (SEM), *n* = 3 biologically independent samples. The data are representative of three experiments with similar results. Statistical analysis was performed using two-tailed unpaired *t*-test. ***P < 0.0001. **B** Comparison of apoptosis induction between vector- and *DUSP16*-transfected HK-1 cells after oxaliplatin or carboplatin treatment by flow cytometry analysis. Bar chart shows the percentage of apoptotic cells (including both AnnexinV⁺7AAD⁺ and AnnexinV⁺7AAD⁻ cells) with mean values ± SEM, *n* = 3 biologically independent samples. The data are representative of three experiments with similar results. Statistical analysis was performed using two-tailed unpaired *t*-test. ***P < 0.0001. **C** Comparison of apoptosis induction between HCT116 and DLD-1 cells in response to oxaliplatin treatment by flow cytometry analysis. Bar chart shows the percentage of apoptotic cells with mean values ± SEM, *n* = biologically independent samples. The data are representative of three experiments with similar results. Statistical analysis was performed using two-tailed unpaired *t*-test. ***P < 0.0001. **D** Comparison of apoptosis induction between vector- and *DUSP16*-transfected HCT116 cells after oxaliplatin or carboplatin treatment by flow cytometry analysis. Bar chart shows the percentage of apoptotic cells with mean values ± SEM, *n* = 3 biologically independent samples. The data are representative of three experiments with similar results. Statistical analysis was performed using two-tailed unpaired *t*-test. ***P < 0.0001. Source data are provided as a Source data file.

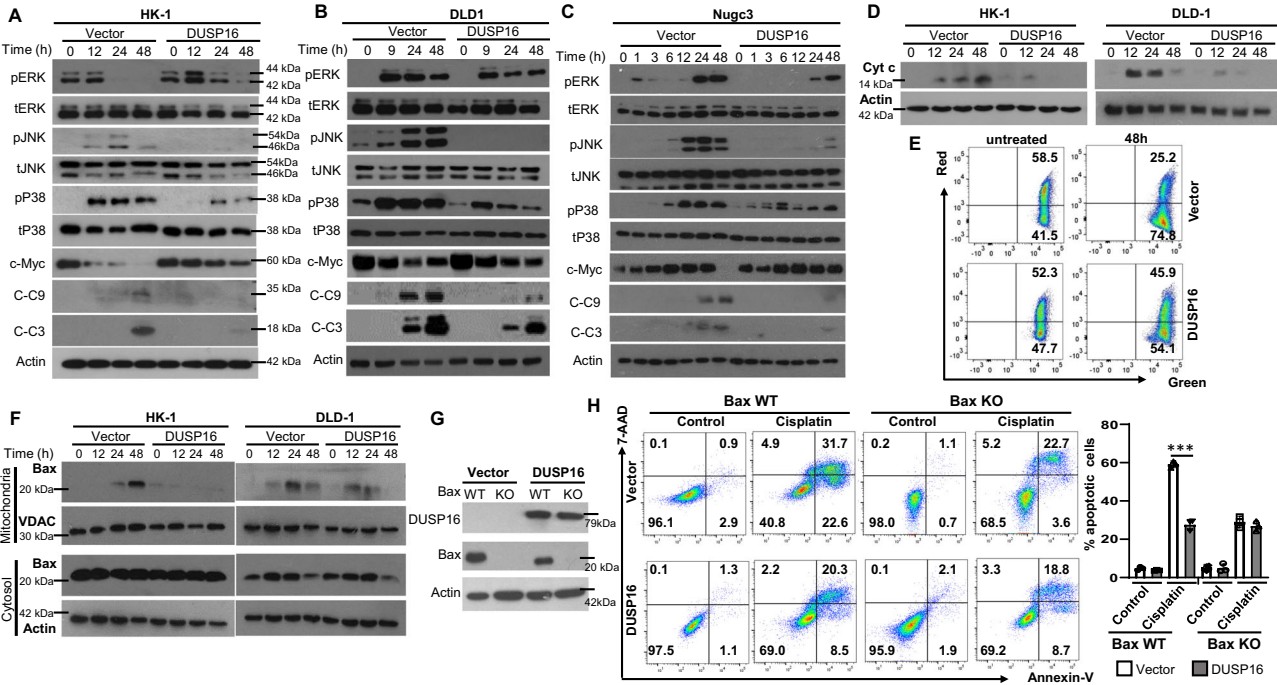

**Fig. 4 *DUSP16* overexpression suppresses BAX-mediated intrinsic apoptotic pathway in various types of cancer cells in response to cisplatin. A–C** Western blot analysis of p-MAPKs, total MAPKs, c-Myc, cleaved caspase 9 (C-C9), and cleaved caspase 3 (C-C3) was carried out on vector- and *DUSP16*-transfected HK-1 (**A**), DLD-1 (**B**), and Nugc3 (**C**) cells after cisplatin treatment. The data is representative of at least two experiments with similar results. **D** Western blot analysis of cisplatin-induced cytochrome *c* (Cyt c) release in cytosolic fractions of vector- and *DUSP16*-transfected HK-1 and DLD-1 cells. The data are representative of three experiments with similar results. **E** Scatterplots depicting changes in the mitochondrial membrane potential in vector- and *DUSP16*-transfected HK-1 cells upon cisplatin treatment. **F** Western blot analysis of BAX expression in mitochondrial and cytosolic fractions of vector- and *DUSP16*-transfected HK-1 and DLD-1 cells. The data are representative of three experiments with similar results. **G** Western blot analysis of *BAX* wild-type (WT) and knockout (KO) DLD-1 cell clones with or without *DUSP16* overexpression. The data are representative of two experiments with similar results. **H** *BAX* WT and KO DLD-1 cells with or without *DUSP16* overexpression were treated with cisplatin for 48 h to examine cell apoptosis by staining with AnnexinV+ and 7AAD+ followed by flow cytometry analysis. Bar chart shows the percentage of apoptotic cells with mean values ± SEM, *n* = 3 biologically independent samples. Statistical analysis was performed using two-tailed unpaired *t*-test. ***P < 0.0001. The data are representative of three experiments with similar results. Source data are provided as a Source data file.

vector- and *DUSP16*-overexpressing cancer cells upon cisplatin treatment was evaluated. In vector-expressing HK-1 cells, constitutive activation of ERK was observed (Fig. 4A). ERK activation was elevated and maintained during the first 12 h, and was subsequently reduced at 24 and 48 h. *DUSP16* overexpression did not affect the initial activation of ERK (Fig. 4A). Although a reduction in activation was observed at 24 and 48 h of treatment, higher levels of ERK activation were detected in the *DUSP16*-expressing cells compared to the vector-expressing cells from 12 h onward. For DLD-1 cells, ERK activation was observed in vector-expressing cells and the activation profile was similar to that of *DUSP16*-expressing cells up until 48 h, when a slight decrease of activation was observed (Fig. 4B). In Nugc3 cells, weak ERK activation was induced by cisplatin at 1 h and onward in response to treatment, and the activation of ERK reached the maximal level at 24 and 48 h, and overexpression of *DUSP16* suppressed the activation of ERK (Fig. 4C). Interestingly, ERK activation was not affected by cisplatin treatment or DUSP16 overexpression in MDA-MB-231 cells (Supplementary Fig. 2D).

Both JNK and p38 were not constitutively activated in HK-1, Nugc3, and MDA-MB-231 cells (Figs. 4A and 4C, and Supplementary Fig. 2D), and were constitutively activated at a low level in DLD-1 cells (Fig. 4B). JNK activation was induced at 12 h after treatment, and reached the maximal activation at 24 h before being subsequently reduced in vector-HK-1 cells (Fig. 4A). In vector-DLD-1 cells, JNK activation was increased steadily with the duration of treatment (Fig. 4B). In vector-Nugc3 cells, JNK

activation was induced and maintained from 6 h until 24 h, followed by reduced activation at 48 h (Fig. 4C), while in vector-MDA-MB-231 cells, JNK activation was induced at 3 h, peaked at 6 h, and maintained up until 24 h, and was then subsequently reduced (Supplementary Fig. 2D). In each of these cell lines, DUSP16 overexpression led to the abrogation of JNK activation by cisplatin treatment (Fig. 4A–C and Supplementary Fig. 2D). Similarly, cisplatin induced or increased the activation of p38, and the activation was maintained for the duration of treatment in vector-expressing HK-1, DLD-1, Nugc3, and MDA-MB-231 cells, and DUSP16 overexpression greatly inhibited p38 activation (Fig. 4A–C and Supplementary Fig. 2D).

Similar results were observed in cells in response to carboplatin or oxaliplatin treatment. DUSP16 overexpression inhibited the activation of JNK, p38, but not ERK in HK-1 cells in response to carboplatin or oxaliplatin treatment (Supplementary Fig. 3C-D), and in DLD-1 and HCT116 cells in response to carboplatin (Supplementary Fig. 3E). These results suggest that DUSP16 regulates cancer cell response to chemotherapy drugs through JNK and p38.

To explore if DUSP16 commonly regulates cancer cell sensitivity to chemotherapeutic agents through JNK and p38, we treated vector- or *DUSP16*-expressing HCT116, HK-1, Nugc3, and MDA-MB-231 cells with 5-Fluorouracil (5-FU) or epirubicin to examine cell apoptosis. Indeed, overexpression of *DUSP16* in all these cells resulted in increased resistance to both 5-FU and epirubicin compared to vector-transfected cells (Supplementary

Fig. 4A-B). Activation of JNK and p38, and expression of cleaved caspase 9 and 3 in *DUSP16*-expressing HCT116 and Nugc3 cells were found to be reduced compared to that in vector-transfected cells in response to 5-FU (Supplementary Fig. 4C-D). These results suggest that DUSP16 more broadly affects cancer cell response to cytotoxic stress through regulation of JNK and p38.

**DUSP16 regulates mitochondria-mediated cell death.** To further understand the regulation of DUSP16 in chemotherapy-mediated cell death, we examined the expression and activation of molecules that are important for cell survival and apoptosis, including c-Myc, caspase 9, and caspase 3. Constitutive expression of c-Myc was observed in HK-1, DLD-1, Nugc3, and MDA-MB-231 (Fig. 4A–C and Supplementary Fig. 2D). Cisplatin treatment resulted in a reduction of c-Myc in HK-1, DLD-1, and MDA-MB-231 cells, but not in Nugc3 cells. Overexpression of *DUSP16* resulted in increased c-Myc expression in HK-1 and MDA-MB-231 cells at 12 h onward upon treatment compared to vector-transfected cells, but not in DLD-1 and Nugc3 cells. These results suggested that c-Myc is not likely responsible for the commonly observed reduction in apoptosis caused by DUSP16 in the four different types of cancer cells. In contrast, reduced expression of cleaved caspase 9 and 3, the active form of these two proteins, were detected in *DUSP16*-expressing cells compared to vector-expressing cells in all four types of cancer cells (Fig. 4A–C and Supplementary Fig. 2D), demonstrating that DUSP16 regulates cisplatin-mediated cell death through inhibiting the activation of caspase 9 and caspase 3. Consistently, reduced cytochrome *c* release was observed in HK-1 and DLD-1 cells with *DUSP16* overexpression compared to vector-expressing cells upon cisplatin treatment (Fig. 4D). Similarly, reduced cleaved caspase 9 and caspase 3 were observed in *DUSP16*-expressing HK-1 cells compared to vector-expressing cells in response to carboplatin or oxaliplatin treatment (Supplementary Fig. 3C-D), and in DLD-1 and HCT116 cells in response to carboplatin (Supplementary Fig. 3E). In addition, reduced cleaved caspase 9 and caspase 3 were observed in *DUSP16*-expressing HCT116 and Nugc3 cells in response to 5-FU and epirubicin treatment (Supplementary Fig. 4C-D). These results suggest that DUSP16 regulates cancer cell apoptosis in response to chemotherapy drugs through the intrinsic or mitochondrial cell death pathway.

Next, we tested the influence of DUSP16 on the mitochondrial membrane potential in response to cisplatin in HK-1 cells using JC-10 staining and flow cytometry. JC-10 aggregates inside the mitochondria and emits orange/red fluorescence. However, upon membrane polarization, JC-10 is disaggregated and forms monomers, resulting in a green emission. The results showed that cisplatin treatment resulted in a reduced JC-10 staining in vector-transfected HK-1 cells, suggesting the decrease of the mitochondrial membrane potential (Fig. 4E). In contrast, *DUSP16*-expressing HK-1 cells had a higher percentage of JC-10 stained cells than vector-transfected cells (45% vs 25%), suggesting that *DUSP16* expression resulted in better mitochondria function in response to cisplatin treatment. Furthermore, we found that cisplatin treatment resulted in increased accumulation of BAX which is a pro-apoptotic protein in the mitochondria, and the accumulation of BAX was reduced by the expression of *DUSP16* in both HK-1 and DLD-1 cells (Fig. 4F). Overall, these results suggest that DUSP16 promotes resistance to chemotherapy drugs in cancer cells through prevention of BAX accumulation in the mitochondria and the activation of mitochondria cell death pathway, thereby suppressing apoptosis.

**DUSP16 regulates mitochondria-mediated cell death through BAX.** To elucidate the role of BAX in DUSP16-mediated cisplatin

resistance, we generated *BAX* knockout DLD-1 cells with or without *DUSP16* overexpression using CRISP/cas9 technology. *BAX* knockout clones with or without *DUSP16* overexpression (one each) were selected to study their response to cisplatin (Fig. 4G). In *BAX*-wild-type (WT)/vector-transfected cells, treatment with cisplatin resulted in 54.3% of apoptotic cells (including AnnexinV⁺7AAD⁻ and AnnexinV⁺7AAD⁺ cells) at 48 h upon treatment (Fig. 4H), whereas that of *BAX* WT/ *DUSP16*-overexpression cells was 28.8% (Fig. 4H). Knockout of *BAX* from vector-transfected cells resulted in a reduction of apoptosis to 26.3%, which was comparable to that of *DUSP16*-overexpression/ *BAX* knockout cells (27.5%), demonstrating that BAX is essential for DUSP16-mediated cisplatin resistance. Consistently, vector-overexpressing/*BAX*-KO, *DUSP16*-overexpressing, and *DUSP16*-overexpressing/*BAX*-KO cells have comparable percentages of live cells (68.5, 69.0, and 69.2%) after cisplatin treatment. These results demonstrate clearly that BAX plays an essential role in DUSP16-mediated cisplatin resistance in DLD-1 cells.

To substantiate the finding on BAX in DUSP16-mediated cisplatin resistance, we treated various types of vector- and DUSP16-overexpressing cancer cells including NPC HK-1, CRC DLD-1, gastric cancer NUGC3, and breast cancer MDA-MB-231 cells with BAX inhibitor Peptide V5. Treatment of HK-1 cells with Peptide V5 successfully inhibited BAX accumulation in mitochondria at 48 h upon cisplatin treatment (Supplementary Fig. 5A). Consequently, great reduction of cell apoptosis was observed in vector-transfected HK-1 cells, but not in *DUSP16*-overexpression cells (Supplementary Fig. 5B). In fact, the percentage of apoptotic cells became comparable between vector- and *DUSP16*-expressing cells. Similar results were observed in vector- and *DUSP16*-expressing DLD-1, NUGC3, and MDA-MB-231 cells (Supplementary Fig. 5C-E). Together, these results further support the essential role of BAX in DUSP16-mediated cisplatin resistance.

**Inhibition of both JNK and p38 in NPC results in reduced apoptosis in response to cisplatin treatment.** Overexpression of *DUSP16* in cancer cells suppressed chemotherapy drug-mediated cell death, which was associated with inhibition of JNK/p38 activation and BAX-mediated apoptosis (Figs. 2–4 and Supplementary Figs. 2–5), suggesting that DUSP16 regulates BAX-mediated apoptosis in cancer cells in response to the treatment via JNK, p38, or both. To confirm the regulation of JNK and p38 by DUSP16, we first transfected Flag-tagged *DUSP16* expression plasmids into HEK293 T cells followed by immunoprecipitation to examine its interaction with JNK/p38 and BAX. The results showed that when DUSP16 was pulled down, endogenous JNK and p38, but not BAX, came down together (Supplementary Fig. 6A), confirming the direct interaction between DUSP16 and JNK/p38, but not BAX. To substantiate the regulation of JNK and p38 activation by DUSP16, we isolated phosphorylated JNK1, JNK2, and p38 from HEK293 T cells stimulated with EGF and incubated with recombinant DUSP16 protein to perform in vitro dephosphorylation assay. It is found that DUSP16 is able to dephosphorylate JNK1, JNK1, and p38 (Supplementary Fig. 6B). Together, these results demonstrate that DUSP16 directly interacts with and inhibits the activation of both JNK and p38, thereby regulating the response to chemotherapy drugs in various types of cancer cells.

To further address the function of JNK and p38 in DUSP16-mediated response to chemotherapy drugs, we treated HK-1 cells with the JNK-specific inhibitor, SP600125, the p38-specific inhibitor, SB203580, or both combined, together with cisplatin. Treatment with SP600125 alone blocked cisplatin-induced JNK

activation (Supplementary Fig. 6C). SB203580 selectively blocks the p38 catalytic activity and the activation of downstream substrates such as the activating transcription factor 2 (ATF2)[65,66]. Consistently, the activation of ATF2 was suppressed by SB203580 in HK-1 cells upon treatment (Supplementary Fig. 6D). Combination of SP600125 and SB203580 resulted in the inhibition of both JNK and p38 activation in response to cisplatin (Supplementary Fig. 6E). These results demonstrate that the activation of JNK or p38 induced by cisplatin is successfully inhibited by their specific inhibitors. Interestingly, inhibition of both JNK and p38 resulted in sustained ERK activation (Supplementary Fig. 6E), mirroring the sustained ERK activation in *DUSP16*-overexpressing HK-1 cells in response to cisplatin treatment (Fig. 4A).

Assessment of HK-1 cell death in cells co-treated with cisplatin and SP600125, SB203580, or both was conducted. Inhibition of JNK or p38 activation resulted in reduced apoptosis after 48 h of cisplatin treatment (AnnexinV/7AAD$^+$ cells—vehicle vs SB203580: 37.4% vs 32.6%, respectively, $p < 0.05$; vehicle vs SP600125: 37.4% vs 31.6%, respectively, $p < 0.01$, Supplementary Fig. 6F). The percentage of live cells, on the other hand, was significantly higher after inhibition of JNK or p38 (Unstained cells—vehicle vs SB203580: 29.8% vs 42.0%, respectively, $p < 0.01$; vehicle vs SP600125: 29.8% vs 38.9%, respectively, $p < 0.01$). Inhibition of both JNK and p38 activation led to a greater reduction in apoptotic cells (AnnexinV/7AAD$^+$ cells—vehicle vs SB203580 + SP600125: 37.4% vs 21.9%, respectively, $p < 0.01$, Supplementary Fig. 6F) and an increased percentage of live cells than those in cells with inhibition of JNK or p38 alone (Unstained cells—vehicle vs SB203580 + SP600125: 29.8% vs 53.0%, respectively, $p < 0.01$) after 48 h of cisplatin treatment. These results suggest that both JNK and p38 promote apoptosis in NPC cells in response to cisplatin and inhibition of both JNK and p38 results in cisplatin resistance. Therefore, DUSP16 promotes cisplatin resistance via inhibition of JNK and p38. Furthermore, sustained ERK activation observed in DUSP16-overexpressing cells or HK-1 parental cells with inhibition of JNK and p38 suggests that DUSP16 regulates the cross-talk between the ERK and JNK/p38 pathways in response to cisplatin.

Next, we examined the expression of cleaved caspase 3 and c-Myc in cisplatin-treated HK-1 cells upon inhibition of JNK, p38, or both. Inhibition of JNK or p38 resulted in reduced levels of cleaved caspase 3 induced by cisplatin compared to cells treated with vehicle, with greater inhibition of cleaved caspase 3 in cells with JNK inhibition than with p38 inhibition (Supplementary Fig. 6G). Inhibition of both JNK and p38 completely blocked the expression of cleaved caspase 3, suggesting that cisplatin-induced caspase 3 activation in NPC requires both JNK and p38. Interestingly, reduced expression of c-Myc was also observed in cells treated with inhibitors. Together, these results suggest that both JNK and p38 contribute to cisplatin-mediated apoptosis by regulating mitochondria-mediated apoptotic pathway.

**DUSP16 promotes resistance to chemotherapy-mediated apoptosis in vivo**. To validate the role of DUSP16 in cancer cell response to chemotherapy in vivo, NOD SCID gamma (NSG) mice were inoculated with either vector-expressing, *DUSP16*-expressing HK-1, or *DUSP16*-expressing DLD-1 cells. Cisplatin treatment (3 mg/kg) was initiated when the size of the tumor reached 0.125 cm$^3$ within a period of 3 weeks. At the end of treatment, tumors were harvested from the mice and examined. For HK-1 cells, the weights of vector- and *DUSP16*-expressing xenografts were not significantly different without treatment (Fig. 5A). Cisplatin treatment resulted in reduced weight of both xenografts, however, this was only statistically significant for

vector-expressing samples, but not for *DUSP16*-expressing ones. In addition, the average weights of the vector-expressing cell xenografts were significantly lower than those of *DUSP16*-expressing ones after cisplatin treatment (Fig. 5A). A similar role for DUSP16 was observed in DLD-1 cells. A significant reduction of tumor weights was only observed in vector-expressing, but not in *DUSP16*-expressing cells upon cisplatin treatment (Fig. 5B), and the average weight of vector-expressing tumors was lower than that of *DUSP16*-expressing tumors. These results support that DUSP16 potentiates tumor resistance to cisplatin in vivo.

Quantification of apoptosis and cell proliferation in HK-1 xenografts was done using immunohistochemical staining with antibodies specific for cleaved caspase 3 and Ki67, respectively. Without cisplatin treatment, there was no significant difference in cleaved caspase 3 staining between vector- and *DUSP16*-expressing xenografts (Fig. 5C, D). Upon cisplatin treatment, the vector-expressing xenografts displayed sixfold greater cleaved caspase 3 staining than the untreated ones. On the other hand, cisplatin treatment resulted in a threefold increase of cleaved caspase 3 staining in *DUSP16*-expressing xenografts (Fig. 5C, D). These results demonstrate the successful induction of apoptosis by cisplatin in both types of tumors, and a twofold reduction of caspase 3 activation was observed in *DUSP16*-expressing xenografts in response to cisplatin (Fig. 5C, D), indicating a reduction in apoptosis. Conversely, Ki67 staining results demonstrated an increase in the number of proliferative cells in *DUSP16*-expressing xenografts treated with cisplatin compared to that in vector-expressing ones (Fig. 5E, F). Comparable Ki67 staining was observed in vector- and *DUSP16*-expressing xenografts without treatment. Cisplatin treatment resulted in a 3.3-fold reduction of Ki67 staining in vector-expressing xenografts, and this Ki67 staining was 3.5-fold and 2.8-fold lower than those of untreated *DUSP16*-expressing and cisplatin-treated *DUSP16*-expressing xenografts, respectively. In contrast, the levels of Ki67 staining in the cisplatin-treated *DUSP16*-expression xenografts were not significantly different from those in the untreated vector-expressing or untreated *DUSP16*-expressing xenografts (Fig. 5E, F). These results further demonstrate that *DUSP16* expression in NPC cells promotes resistance to chemotherapy in vivo.

**Knockdown of DUSP16 expression sensitizes NPC to chemotherapy in vitro and in vivo**. To validate the role of DUSP16 in cisplatin-mediated responses, we performed knockdown of DUSP16 using the CRISPR/Cas9 system in undifferentiated NPC C666-1 cells which express higher levels of DUSP16 and are more resistant to cisplatin-mediated apoptosis compared to HK-1 cells (Fig. 1). CRISPR/Cas9 constructs targeting the *DUSP16* gene or control plasmids were transfected into C666-1 cells. Reduced expression of *DUSP16* was observed at both mRNA and protein levels in cells transfected with the *DUSP16* CRISPR/Cas9 construct (KD) as compared to cells transfected with the vector control (Vector) (Fig. 6A). To test the effect of *DUSP16* knockdown (KD) on cellular response to cisplatin, Vector or KD cells were treated with 30 µg/mL of cisplatin to assess apoptosis. We found that cisplatin treatment led to a significant increase in apoptosis in the *DUSP16*-KD cells compared to the Vector cells (Vector vs. KD; 21.4% vs. 29.2%) (Fig. 6B, C). There was also a lower number of surviving KD cells than control cells after cisplatin treatment (Vector vs. KD; 76.1% vs. 62.5%). These results demonstrate that *DUSP16* expression in C666-1 cells promotes cisplatin resistance, confirming that DUSP16 inhibits cisplatin-mediated apoptosis in NPC.

The activation of the MAPKs was also examined in the KD and Vector cells after cisplatin treatment. Increased JNK activation

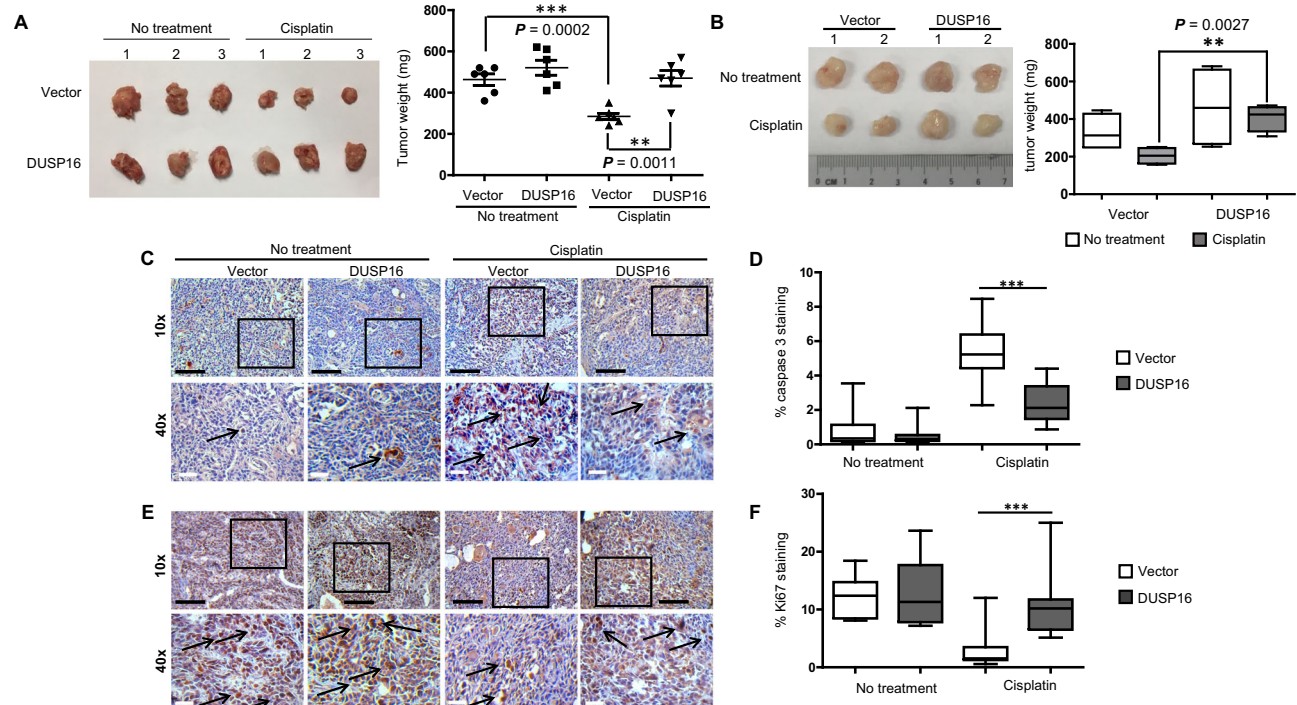

**Fig. 5 Overexpression of *DUSP16* resulted in increased resistance of cancer cells to cisplatin treatment in vivo. A** Vector- or *DUSP16*-transfected HK-1 cells were inoculated into NSG mice. Cisplatin was injected at 3 mg/kg body weight every 3 days. After 20 days, tumors were harvested to measure the sizes and weights. The data are expressed as dots representing the weight of each tumor; the scale bars are the mean ± standard error of the mean (SEM) of tumor weights from six mice in each treatment group of the representative experiment (mean ± SEM, $n = 6$). Statistical analysis was performed using two-tailed unpaired *t*-test. **B** Growth of untreated and cisplatin-treated xenografts of vector- and *DUSP16*-transfected DLD-1 cells. Box-and-whisker plots show means ± SEM of tumor weights from four mice from each treatment group per experiment (mean ± SEM, $n = 4$). Boxes correspond to the 25th, 50th/ median and 75th percentiles; whiskers denote maximum and minimum. Statistical analysis was performed using two-tailed unpaired *t*-test. **C, D** IHC analysis for cleaved caspase 3 of HK-1 tumor sections; representative images of cleaved caspase 3 staining ($n = 6$) were shown (**C**). Boxed regions were magnified and presented under the respective panels. Black arrows highlight the caspase 3 staining. Box-and-whisker plots show means ± SEM of the means of cleaved caspase 3 staining in each group of mice (**D**). Boxes correspond to the 25th, 50th/median and 75th percentiles; whiskers denote maximum and minimum. Statistical analysis was performed using two-tailed unpaired *t*-test. ***$P < 0.0001$ (mean ± SEM, $n = 3$). **E, F** IHC analysis for Ki67 of tumor sections. Representative images of Ki67 staining were shown (**E**). Boxed regions were magnified and presented under the respective panels. Black arrows highlighted the Ki67 staining. Box-and-whisker plots show means ± SEM of Ki67 staining (**F**) from each group of mice ***$P < 0.0001$ (mean ± SEM, $n = 3$). The data are representative of three experiments with similar results. Source data are provided as a Source data file.

and earlier induction of phosphorylated p38 in the *DUSP16*-KD cells compared to control cells were observed after 24 and 48 h of treatment (Fig. 6D). Interestingly, earlier induction of ERK activation was also noted in the *DUSP16*-KD cells compared to the Vector cells (Fig. 6D). Furthermore, the expression of c-Myc was reduced in the KD cells compared to that in control cells after treatment (Fig. 6D). These results are in line with our observations in HK-1 cells, where overexpression of *DUSP16* resulted in inhibited JNK and p38 activation, increased c-Myc expression, and reduced cell apoptosis (Figs. 3B and 4A), suggesting that DUSP16 negatively regulates cisplatin-mediated apoptosis in NPC cells by regulating the MAPK pathways, namely JNK and p38.

To examine the function of DUSP16 in C666-1 cells in response to cisplatin in vivo, *DUSP16*-KD and Vector cells were inoculated into NSG mice. Tumor growth was assessed with or without 3 mg/kg of cisplatin treatment. We found that the weights of the untreated Vector and *DUSP16*-KD xenografts did not differ significantly (Fig. 7A). The difference of the tumor weights in Vector C666-1 cells before and after treatment was also not significant. However, the average weight of the cisplatin-treated Vector xenografts was significantly larger (1.68-fold) than the cisplatin-treated *DUSP16*-KD xenografts (Fig. 7A), indicating that *DUSP16*-KD C666-1 tumors were more sensitive to growth

inhibition by cisplatin than Vector tumors. Cleaved caspase 3 staining of the tumor sections demonstrated that *DUSP16*-KD xenografts had 1.4-fold greater cleaved caspase 3 staining compared to that of Vector xenografts in response to cisplatin (Fig. 7B), indicating a greater induction of apoptosis. On the other hand, the cisplatin-treated Vector xenografts showed 4.88-fold greater Ki67 staining than that of the cisplatin-treated *DUSP16*-KD xenografts (Fig. 7C). These results further demonstrate that DUSP16 promotes cisplatin resistance in NPC.

**Levels of DUSP16 expression in head and neck squamous cell carcinoma (HNSCC) patients is associated with patient survival probability.** To test our hypothesis that strong DUSP16 expression is associated with poor survival in a patient population, we performed clinicopathological analysis on locally advanced squamous cell carcinoma (SCC) of the head and neck (HNSCC) cohort where concurrent single-agent cisplatin treatment with radiotherapy is the standard first-line regime. The characteristics of the HNSCC cohort are summarized in Supplementary Table 1. DUSP16 expression was determined by immunohistochemistry (Fig. 7C), with the disease-free survival (DFS)—defined as the interval between the completion of chemoradiation to the first histologically or radiologically confirmed

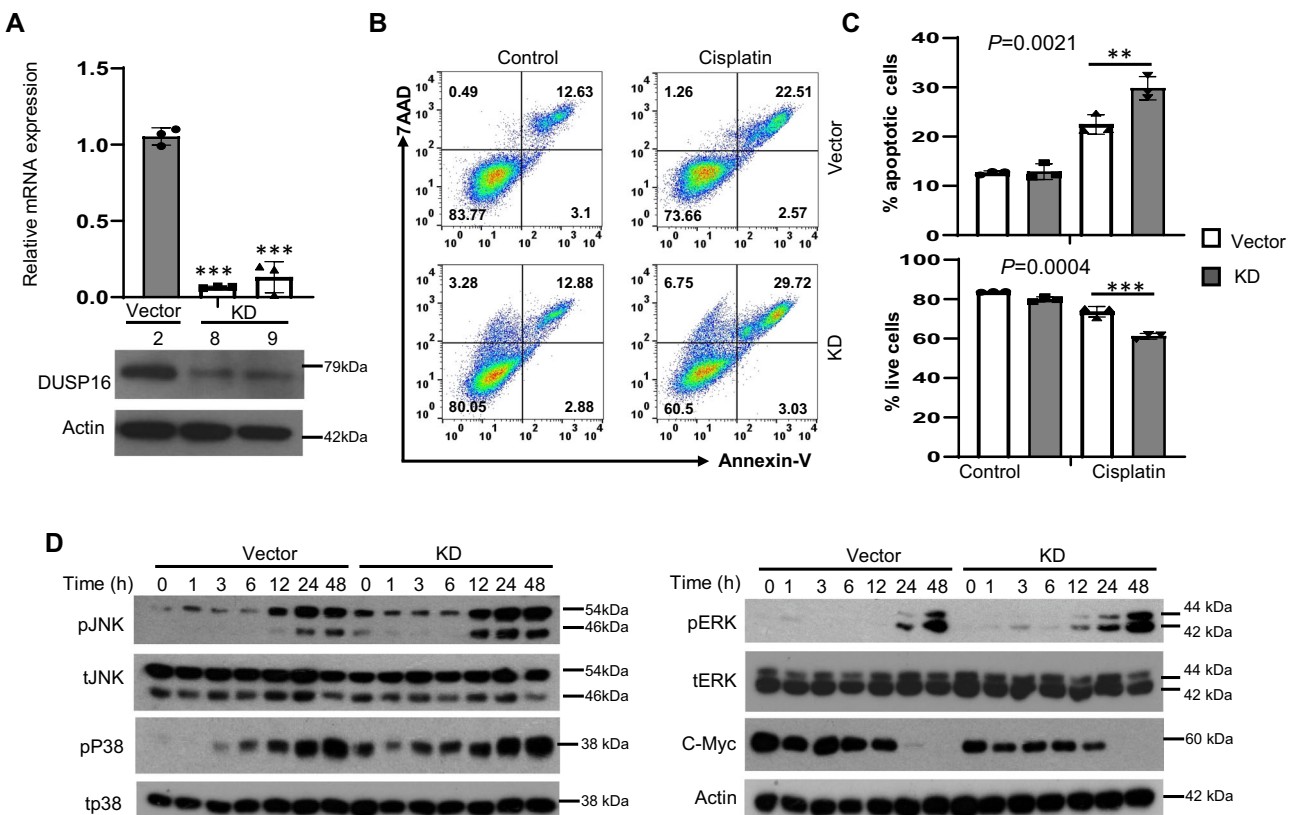

**Fig. 6 *DUSP16* knockdown enhances apoptosis in C666-1 cells in response to cisplatin.** C666-1 cells were transfected with *DUSP16* CRISPR/Cas9 constructs to knockdown *DUSP16* expression. **A** Bar chart shows the mRNA expression in vector and *DUSP16* construct transfected cells with mean values ± SEM, *n* = 3 biologically independent samples. Statistical analysis was performed using two-tailed unpaired *t*-test. Protein levels of DUSP16 were quantified by immunoblotting. ***P < 0.0001. **B** Apoptosis of control and *DUSP16* knockdown (KD) cells after cisplatin treatment was measured by AnnexinV/7AAD staining and flow cytometry. **C** Bar charts show the percentage of apoptotic and surviving (unstained) cells with mean values ± SEM, *n* = 3 biologically independent samples. Statistical analysis was performed using two-tailed unpaired *t*-test. The data are representative of three experiments with similar results. **D** Western blot analysis of the pJNK, pP38, pERK, their total protein levels, and c-Myc in control or DUSP16-KD C666-1 cells was carried out after cisplatin treatment. The data are representative of three experiments with similar results. Source data are provided as a Source data file.

tumor recurrence locally in the head and neck region or distant metastases—was defined in 50 patients. A shorter DFS was determined for patients with higher expression of DUSP16 (log-rank test, *p* = 0.042), suggesting that there may be an association between higher DUSP16 expression and poor survival of HNSCC patients (Fig. 7D). Human papillomavirus (HPV) is a known cause of HNSCC and is associated with better prognosis. Using p16 as a surrogate biomarker for HPV infection, we showed that both the DUSP16-high (19%) and -low (18%) cohorts have identical HPV-positive cases (Supplementary Table 1), indicating that HPV status is not a confounder in the association of DUSP16 and DFS among the HNSCC cases.

**High DUSP16 protein expression in breast cancer patients receiving platinum drug treatment is associated with reduced survival probability.** To substantiate the role of DUSP16 in the regulation of cancer cell sensitivity to platinum-based chemotherapy, we examined a cohort of breast cancer patients who were treated with platinum-based drugs including cisplatin, carboplatin, and lobaplatin after surgery. Within the 113 breast cancer patients, there were 72 triple-negative breast cancer (TNBC), 25 HER2 positive breast cancer, and 16 luminal breast cancer. DUSP16 protein expression in these breast tumors was determined by immunohistochemistry (Fig. 7E). Analysis of DFS in association with DUSP16 expression levels demonstrated that

patients (*n* = 45) with high DUSP16 protein expression in their tumors had significantly reduced survival probability compared to patients (*n* = 68) with low DUSP16 (*p* < 0.0001) (Fig. 7F). These results provide further evidence on the role of DUSP16 in regulation of cancer patient sensitivity to chemotherapy.

## Discussion

Chemotherapeutic agents, such as cisplatin, remain the backbone of treatment for various solid tumors including NPC and gastric cancer, but the development of resistance is a major cause of treatment failure. Understanding the mechanisms underlying development of resistance and identification of biomarkers of resistance will be critical for overcoming this limitation and for the development of effective therapy. The MAPK pathway has been shown to be important in mediating responses to cisplatin treatment in various cancers. As major negative regulators of the MAPKs, altered expression of DUSPs/MKPs may influence the outcome of chemotherapy in cancers. While numerous studies have shown that loss of DUSP expression is correlated with progression of several cancers[67,68], others have associated the gain of DUSP expression with cancer progression, drug resistance, and poor prognosis[67,68]. This study examines the regulatory function of one DUSP member, DUSP16, in NPC, colorectal cancer, gastric cancer, and breast cancer in response to chemotherapy drug treatment. Given that resistance to chemotherapeutic agents is a

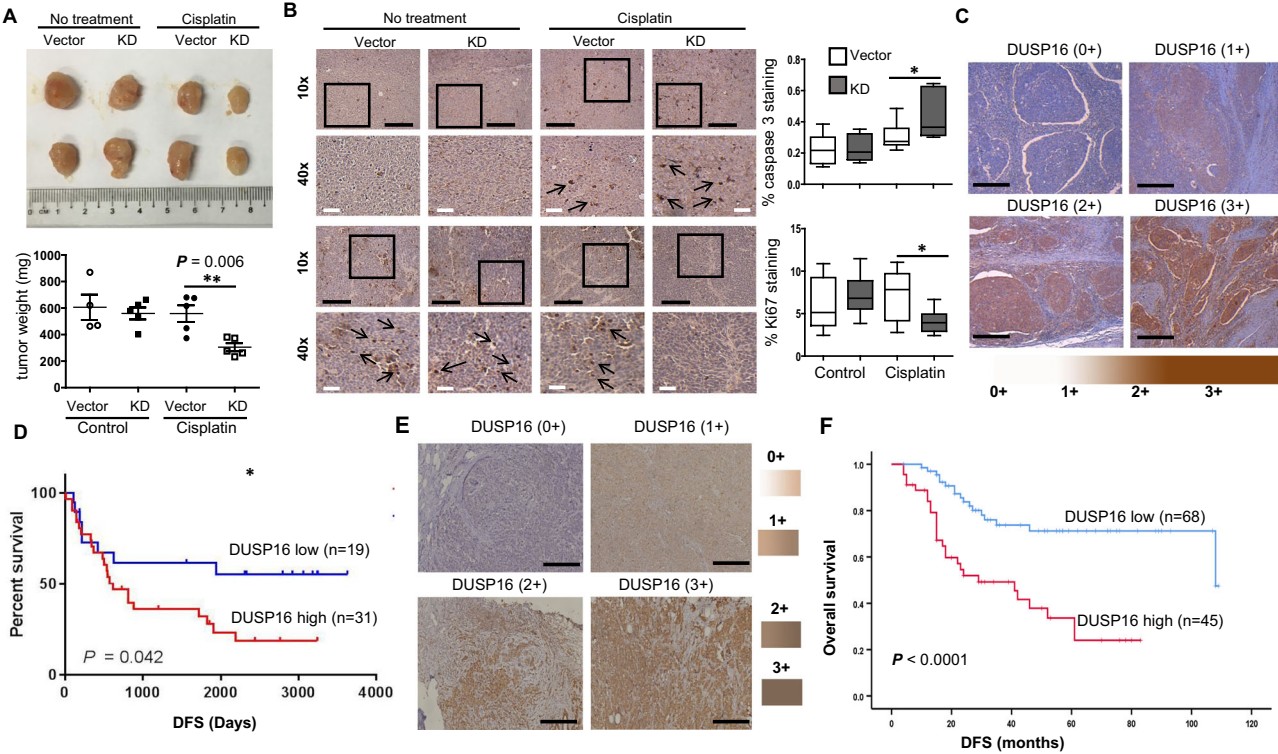

**Fig. 7 DUSP16 knockdown in C666-1 cells resulted in increased sensitivity to cisplatin in vivo and DUSP16 levels were inversely associated with head and neck squamous cell carcinoma (HNSCC) patient and breast cancer patient survival. A** Mice were inoculated with control or *DUSP16-KD* cells. Cisplatin was injected at 3 mg/kg body weight every 3 days. After 20 days, tumors were harvested, and the sizes and weights were measured. The data are expressed as dots representing the weight of each tumor; the scale bars are the mean ± SEM of tumor weights from 4 (the vector control group) or 5 mice (the other groups) of the representative experiment. Statistical analysis was performed using two-tailed unpaired *t*-test. **B** Representative images of immunohistochemistry analysis for cleaved caspase 3 and Ki67 staining of tumor sections were shown. Boxed regions were magnified and presented under the respective panels. Black arrows highlight the caspase 3 and Ki67 staining, respectively. Box-and-whisker plots show mean values ± SEM of cleaved caspase 3 and Ki67 staining from each group of mice (*n* = 4). Boxes correspond to the 25th, 50th/median and 75th percentiles; whiskers denote maximum and minimum. Black scale bars = 200 μm, white scale bars = 50 μm. Statistical analysis was performed using two-tailed unpaired *t*-test. \**P* < 0.05. The data are representative of three experiments with similar results. **C** Representative images of HNSCC specimens (*n* = 50) showing negative (0 +), low (1 +), intermediate (2+), and high (3+) staining of DUSP16 in tumor cells. Black scale bars = 200 μm. **D** Patients with higher DUSP16 expression showed lower disease-free survival (DFS) compared to patients with low DUSP16 expression (log-rank test, two-sided, *P* = 0.042). **E** Representative images of breast cancer specimens (*n* = 113) showing negative (0 +), low (1 +), and high (3 +) staining of DUSP16 in tumor cells. Black scale bars = 200 μm. **F** Among the 113 breast cancer patients (age 28–64) treated with cisplatin, carboplatin, or lobaplatin after surgery, high DUSP16 protein expression in their tumors was associated with lower disease-free survival compared to patients with low DUSP16 expression (log-rank test, two-sided, *P* < 0.0001). Source data are provided as a Source data file.

major drawback in cancer therapy, information from this study may aid in understanding the mechanisms behind resistance to this group of drugs. In addition, the role of DUSP16 in the response of cancers to chemotherapeutic agents identified in this study suggests that DUSP16 could be a marker for predicting the efficacy of chemotherapy, and also a target for the development of therapies to improve treatment efficacy.

First, we discovered that DUSP16 was differentially expressed in NPC, CRC, and gastric cancer cell lines. The undifferentiated EBV-harboring NPC cell line, C666-1, expressed a higher level of DUSP16 than the three differentiated EBV-negative NPC cell lines (HK-1, HONE-1, CNE-1) (Fig. 1A, B). It is possible that the differentiation status, the presence of EBV, or both may be factors in the differential expression of DUSP16 in different types of NPC, which is not covered in this study and warrants further investigation. Nevertheless, we found that the DUSP16-higher C666-1 cells are more resistant to cisplatin-mediated cell death than the HK-1 cells which have minimal DUSP16 expression (Fig. 1D). Furthermore, C666-1 cells are more resistant to carboplatin than HK-1 cells (Fig. 3A), suggesting the possible involvement of this molecule in NPC response to chemotherapy

drugs. Similarly, the DUSP16-higher CRC cell line DLD-1 and gastric cancer cell line Ags showed more resistance to the three chemotherapy drugs than DUSP16-lower HCT116 and Nugc3, respectively (Fig. 1E, F, Supplementary Fig. 1C-D and Fig. 3C). These results suggest that DUSP16 may have similar function in NPC, CRC, and gastric cancer in response to various chemotherapeutic agents and the level of DUSP16 could serve as a marker for sensitivity to chemotherapy in these cancers.

To further elucidate the function of DUSP16 in cancer response to chemotherapy, we overexpressed this molecule in the NPC cell line HK-1, the CRC cell line DLD-1, and HCT116, the gastric cancer cell line Nugc3 as well as the breast cancer cell line MDA-MB-231. A CRISPR/Cas9-mediated approach was also used to knock down the expression of *DUSP16* in C666-1 NPC cells. Overexpression of *DUSP16* in HK-1, DLD-1, Nugc3, and MDA-MB-231 cells impaired their sensitivity to cisplatin, carboplatin, oxaliplatin, 5-FU, and epirubicin whereas *DUSP16* knockdown in C666-1 cells sensitized them to cisplatin in vitro (Figs. 2, 3, 6 and Supplementary Figs. 2–4). These trends were also observed in in vivo HK-1, DLD-1, and C666-1 xenograft mouse models (Figs. 5 and 7). In addition, high DUSP16

expression was found to be associated with poorer disease-free survival in a cohort of HNSCC patients upon completion of chemoradiation therapy (Fig. 7C, D) and in a cohort of breast cancer patients who received platinum-based chemotherapy after surgery (Fig. 7E, F). These observations confirm the role of DUSP16 in promoting resistance to chemotherapy drugs in NPC, CRC, gastric, and breast cancer.

The impaired sensitivity to chemotherapy drugs in *DUSP16*-expressing cancer cells was associated with inhibited activation of JNK and p38 (Fig. 4 and Supplementary Figs. 2, 3, 5), suggesting that JNK and p38 activation in these cancer cells promotes cell apoptosis in response to chemotherapy treatment. We confirmed the direct interaction between DUSP16 and JNK/p38 and desphophorylation of JNK/p38 by DUSP16 (Supplementary Fig. 6A-B). The cell death promoting functions of JNK and p38 were confirmed by studies using a JNK-specific inhibitor, a p38-specific inhibitor, or both combined (Supplementary Fig. 6). Furthermore, knockdown of *DUSP16* in C666-1 cells resulted in enhanced JNK and p38 activation associated with increased cell apoptosis upon cisplatin treatment (Fig. 6). Therefore, JNK and p38 activation by chemotherapeutic compounds promotes cell apoptosis, and DUSP16 expression promotes resistance to these compounds in cancers by inhibiting both JNK and p38.

Chemotherapy drugs such as cisplatin induce the activation of JNK and p38 in various types of cancer cells, and their activation was found to be critical in determining the cellular response to the drugs. Both JNK and p38 could be either pro- or anti-apoptotic, depending on the type of cells, the nature of the stimulus, the duration of their activation, and their interaction with other factors[64]. The DUSP/MKP proteins primarily target MAPKs to control the magnitude and duration of their activation. Each DUSP/MKP has its specific expression pattern and/or expression regulation among various tissues/organs and cell types, specific subcellular localization, and substrate preferences[67]. In this study, prolonged activation of JNK and p38 in NPC HK-1 and C666-1 cells, and CRC DLD-1 cells, gastric cancer Nugc3 cells, and breast MDA-MB-231 cells in response to chemotherapeutic agents was observed and DUSP16 expression inhibited both JNK and p38 activation in all the cell lines (Fig. 4A–C and Supplementary Figs. 2D, 3, 4, 6), which was associated with reduced apoptosis (Fig. 2 and Supplementary Figs. 2–5). Conversely, knockdown of *DUSP16* in C666-1 resulted in increased JNK and p38 activation and increased apoptosis (Fig. 6D). Interestingly, C666-1 cells, which constitutively express a higher level of DUSP16, are intrinsically more resistant to cisplatin than HK-1 cells (Fig. 1D). A similar phenomenon was observed in the CRC cell lines DLD-1 and HCT116 (Fig. 1E and Supplementary Fig. 1C), and the gastric cancer cell lines Ags and Nugc3 (Fig. 1F and Supplementary Fig. 1D). These results demonstrate that prolonged JNK and p38 activation is pro-apoptotic in NPC, CRC, and gastric cancer, and by inactivating JNK and p38, DUSP16 is part of the strategy employed by these cancer cells in intrinsic or acquired cisplatin resistance.

Interestingly, inhibition of JNK and p38 activation by DUSP16 was associated with prolonged ERK activation in response to cisplatin treatment only in HK-1 cells (Fig. 4A), but not in DLD-1, Nugc3 (Fig. 4B, C), or MDA-MB-231 cells (Supplementary Fig. 2D). Similarly, inhibition of both JNK and p38 using their specific inhibitors also resulted in prolonged ERK activation in HK-1 cells (Supplementary Fig. 6E). These observations suggest a cross-talk between the JNK/p38 and ERK pathways in NPC in response to cisplatin treatment. It is unclear why overexpression of DUSP16 resulted in a cross-talk between JNK/p38 and ERK only in NPC cells, but not in CRC, gastric cancer cells, or breast cancer cells. It is possible that inhibiting the activation of JNK/p38 via DUSP16 or specific inhibitors resulted in a compensatory enhancement of ERK activation due to a reduced expression of ERK-specific phosphatases in HK-1 cells, but not in other types of cancer cells. However, this requires further investigation. Our data suggest that ERK activation in responses to chemotherapeutic drugs may differ depending on the type of cancer cells, and ERK is not likely to be involved in the DUSP16-mediated resistance commonly observed in various types of cancer cells.

Apoptosis is a frequent mechanism of drug-induced tumor cell death, and alterations in apoptotic pathways are some of the most important mechanisms underlying the development of chemoresistance of cancer cells[64]. The DNA damage-induced apoptosis process begins with the translocation of the pro-apoptotic protein BAX from the cytosol to the mitochondria, triggering a cascade of events in the mitochondria leading to the release of cytochrome *c*. Release of cytochrome *c* activates the caspase 9–caspase 3 pathway. Activated caspase 3, an apoptosis executioner in both the intrinsic and the extrinsic pathway, is largely responsible for cleavage of many cellular proteins, leading to the morphological manifestations of apoptosis[69,70]. Cancer cells have evolved multiple mechanisms such as expressing high levels of anti-apoptotic factors or inactivating pro-apoptotic molecules to escape the induction of apoptosis[71]. This study demonstrates that inducing the expression of DUSP16 may be a common strategy of various solid tumors, including NPC, CRC, gastric cancer, and breast cancer, to evade chemotherapy-induced cell apoptosis. We showed that the activation of caspase 9 and caspase 3 was reduced in *DUSP16*-expressing NPC HK-1, CRC DLD-1, gastric cancer Nugc3, and breast cancer MDA-MB-231 cells (Fig. 4A–D, Supplementary Figs. 2D, 3C-E, and 4C-D), which was associated with reduced cytochrome *c* release in response to chemotherapeutic agents. DUSP16-mediated reduction of caspase 3 activation was further confirmed in *DUSP16*-expression HK-1 xenografts (Fig. 5C, D); whereas increased caspase 3 activation was detected in *DUSP16* knockdown C666-1 tumors (Fig. 7B). In addition, the combination of specific inhibitors for JNK and p38 together were able to suppress caspase 3 activation, and were more efficient in inhibiting caspase 3 activation than when used individually (Supplementary Fig. 6E), suggesting that DUSP16 regulates caspase 3 through JNK and p38. Accumulation of BAX protein in mitochondria was found to be reduced in both HK-1 and DLD-1 cells with DUSP16 overexpression in response to cisplatin treatment (Fig. 4F). These data suggest that caspase 3 is a downstream activator of JNK/p38 upon cisplatin treatment, and can be targeted by DUSP16 through inactivation of JNK and p38. Examination of BAX protein in mitochondria isolated from HK-1 and DLD-1 cells demonstrated that cisplatin treatment induced BAX accumulation in mitochondria and *DUSP16* overexpressing inhibited its accumulation (Fig. 4F). Both JNK and p38 have been shown to promote BAX translocation to mitochondria to mediate apoptosis in response to drug treatment in cancer cells[64,72]. Studies using *BAX* knockout cells with or without *DUSP16* overexpression provided definitive evidence on the essential role of BAX in DUSP16-mediated chemoresistance (Fig. 4G, H). Therefore, DUSP16 regulates JNK/p38-BAX signaling pathway to mediated resistance to various chemotherapeutic agents in cancer (Supplementary Fig. 7). It is believed that MAPKs, such as JNK, are a major factor deciding the fate of a cell in response to chemodrugs such as cisplatin[64]. However, the need for JNK and p38 in normal cell maintenance poses problems for their inhibitors when used for therapy. It is plausible that DUSP16, whose expression is induced by chemotherapeutic agents, targets both JNK and p38 to regulate cancer cell response, and therefore, might be an important target for enhancing the efficacy of chemotherapy.

We have analyzed the association of DUSP16 with patient survival in two cohorts of cancer patients, a small cohort of

HNSCC patients and a cohort of 113 breast cancer patients, and found that high DUSP16 levels in tumors were associated with significantly lower patient survival. Future studies could be done to characterize DUSP16 expression levels and to correlate the status of JNK/p38 activation in the tumor tissues with the adjacent normal regions in chemosensitive and chemoresistant cancer patients. This would be useful in verifying the findings of our study and in determining treatment strategies for these groups of patients.

In summary, our study demonstrates that DUSP16 targets JNK and p38 to regulate the common cell death pathway, the BAX/caspase 3–caspase 9 pathway, in various types of solid tumors, including NPC, CRC, gastric cancer, and breast cancer to regulate chemotherapy-induced apoptosis. The important function of DUSP16 in mediating sensitivity to chemotherapy in various types of cancers suggests that it could serve as a predictive marker for stratifying responsive patients, and as a target for the development of therapies to improve the effectiveness of chemotherapy.

## Methods

**Cell culture**. NPC cell lines, C666-1 and HK-1, were kindly provided by Professor Chan Soh Ha (NUS Immunology Center) and were maintained in RPM1 1640 (Gibco, USA) supplemented with 10% fetal bovine serum (FBS) in the presence of penicillin (100 IU/mL) and streptomycin (100 μg/mL) at 37 °C under a humidified atmosphere of 95% air and 5% $CO_2$. C666-1 is an undifferentiated NPC cell line which stably maintains the EBV episomes while HK-1 is a well-differentiated, EBV-negative NPC cell line.

DLD-1 and HCT116 colorectal carcinoma cell lines, and MDA-MB-231 breast cancer cell line were obtained from ATCC and maintained in DMEM containing 10% fetal bovine serum (FBS) with penicillin (100 IU/mL) and streptomycin (100 μg/mL) at 37 °C under a humidified atmosphere of 95% air and 5% $CO_2$.

**Cell treatments**. Cisplatin (Sigma-Aldrich) was added to the cell media from concentrated stocks (1 mg/mL) dissolved in sterile 0.9% NaCl/water. The same amount of solvent (sterile 0.9% NaCl/water) was added to the control wells. At the end of each time point, the cells were harvested for subsequent experiments. Floating cells were collected by centrifugation and removal of the supernatant, and combined with the harvested adherent cells for these experiments.

Specific inhibitors for the individual MAPKs (ERK: PD98059, p38: SB203580, or JNK: SP600125, Sigma-Aldrich) were used in the MAPK inhibition studies. A 1-h pre-treatment with inhibitors of ERK (50 μM of PD98059), p38 (20 μM of SB203580), or JNK (20 μM of SP600125) was done prior to cisplatin treatment. DMSO was added to each control well as the control solvent in place of the MAPK inhibitors.

**Establishment of DUSP16-overexpressing cancer cell lines**. A *DUSP16* expression plasmid was made by subcloning human *DUSP16* cDNA to a pPy-CAGIP vector. The pPyCAGIP vector confers resistance to puromycin via a puromycin resistance gene to allow for selection of positively transfected clones. Oligonucleotide primers (Integrated DNA Technologies) for *DUSP16* with XhoI and NotI restriction sites were used to amplify the full-length cDNA of the *DUSP16* gene. The forward primer was synthesized with an XhoI restriction site and the reverse primer with a NotI restriction site (Supplementary Table 2).

Cancer cell lines were transfected with either empty pPyCAGIP vector or with the DUSP16-pPyCAGIP vector using Lipofectamine® LTX (Invitrogen). Transfected cells were selected using media containing puromycin. Selected single cells were isolated and assessed for DUSP16 expression by quantitative real-time PCR (qPCR) and western blot.

***DUSP16* knockout/knockdown by CRISPR/Cas9 transfection**. To generate Bax-KO cells, a pair of short guide-RNA (sgRNAs) were ordered from Integrated DNA Technologies (Supplementary Table 2). Oligonucleotides for guide RNAs were cloned into the pSpCas9(BB)-2A-GFP (PX458) plasmid (Addgene plasmid #48138) as described by Ran et al.[73]. BAX sgRNA plasmids were then transfected into DLD-1 cells using Lipofectamine®3000. 24–48 h after transfection, GFP positive cells were sorted and seeded individually into 96-well plates.

Knockdown of DUSP16 in C666-1 cells was performed by transfection with DUSP16 CRISPR/Cas9 KO Plasmid (sc-405727, Santa Cruz) and DUSP16 HDR plasmid (sc-405727-HDR, Santa Cruz) with UltraCruz® Transfection Reagent (sc-395739) according to the manufacturer's instructions. Cells transfected with Control CRISPR/Cas9 plasmid (sc-418922) were used as control.

**RNA extraction, reverse transcription, and real-time polymerase chain reaction**. Total RNA was isolated from the cells using TRIzol reagent (Invitrogen) according to manufacturer's instructions. One microgram of RNA was reverse transcribed into complementary DNA (cDNA) to be used for qPCR. qPCR was performed using SsoAdvanced SYBR Green supermix and CFX Connect RT System (BioRad) to examine the expression of *DUSP16* with *ACTIN* as internal control using respective primer pairs (Supplementary Table 2).

**Western blot**. Western blot analysis was carried out to assess protein expression levels. Total protein was extracted from the cells by addition of lysis buffer (10 mM Tris, pH 8.0, 120 mM NaCl, 0.5% NP-40, 1 mM EDTA) containing protease inhibitor cocktail (Roche, Switzerland) and phosphatase inhibitor cocktail (Roche). Cultured cells were scraped and whole-cell lysates were prepared by centrifugation at $16,363 \times g$ for 15 min at 4 °C. Equal amounts of protein were loaded alongside a pre-stained protein ladder (1st Base) and electrophoresed on 10% SDS-polyacrylamide gels. The resolved proteins were transferred onto PVDF membranes (Immobilon transfer membrane, Millipore Corporation). After electroblotting, the membranes were blocked with Tris-buffered saline with Tween 20 (1xTBST: 20 mM Tris base, 137 mM NaCl, 0.1% Tween 20) containing 5% non-fat dry milk at room temperature for 1 h before overnight hybridization with appropriate concentrations of primary antibodies (Supplementary Table 3) diluted in blocking buffer. The membranes were then incubated with anti-rabbit HRP-conjugated secondary antibody (NA934, GE Healthcare, UK) for 1 h at room temperature. Antigen-antibody complexed to the blotted proteins were detected using chemiluminescence detection system. The chemiluminescent signal was developed on Amersham Hyperfilm ECL (GE Healthcare).

**AnnexinV/7-(amino-actinimycin D) AAD staining**. Apoptosis was quantified by FITC-conjugated annexin V staining of externalized phosphatidylserine (PS), a reliable marker for early apoptosis[74]. 7AAD was used to quantify dead cells. This staining distinguishes between early apoptotic (annexin V+), late apoptotic or necrotic (annexin V+/7AAD+), and necrotic cells (7AAD+). Equal numbers of NPC cells were grown in 6-well plates ($0.2 \times 10^6$ cells/well) and treated with cisplatin over 24 and 48 h. Adhering and detached cells were collected at the end of treatment and washed with 3% FBS in PBS. Cells were then suspended in 100 μL 1× binding buffer and stained with 5 μL of AnnexinV-FITC and 5 μL of 7AAD (BioLegend) per sample for 15 min in the dark at room temperature. 400 μl binding buffer was then added before cells were analyzed by flow cytometry. Cells that were not stained by annexin V or 7AAD are viable cells, and cells stained by both annexin V and 7AAD are apoptotic cells. The percentages of AnnexinV-FITC/7AAD stained cells were analyzed by Flowjo software.

**Mitochondrial and cytoplasmic isolation**. Isolation of the mitochondrial and cytoplasmic fractions of the cultured cell lines was carried out using the Mitochondria Isolation Kit for Cultured Cells (Pierce) according to the manufacturer's instructions. Briefly, $2 \times 10^7$ cells were incubated with 800 μL of mitochondria isolation reagent A and then homogenized by sequential vortexing and incubation on ice with reagent B. Unlysed cells and large debris were pelleted by centrifugation at $700 \times g$ for 10 min at 4 °C. The supernatant was further centrifuged at $3000 \times g$ for 15 min at 4 °C. The supernatant (cytosolic fraction) was collected, and lysis of the pellet (mitochondrial fraction) was carried out by addition of lysis buffer (10 mM Tris, pH 8.0, 120 mM NaCl, 0.5% NP-40, 1 mM EDTA) containing protease inhibitor cocktail (Roche, Switzerland) and phosphatase inhibitor cocktail (Roche), and by centrifugation at $16,363 \times g$ for 15 min at 4 °C.

**Mitochondrial membrane potential assay**. Changes in mitochondrial membrane potential in HK-1 cell lines after cisplatin treatment were assessed using JC-10 Mitochondrial membrane potential assay kit (Abcam) according to the manufacturer's instructions. NPC cells were grown in 6-well plates ($0.2 \times 10^6$ cells/well) and treated with cisplatin over 48 h. Cells were collected at the end of treatment and washed with 3% FBS in PBS. Samples were stained for 30 min in the dark at room temperature before analysis by flow cytometry. The percentages of green (apoptotic/necrotic) and orange/red (normal) cells were analyzed by Flowjo software.

**Assessment of in vivo tumor growth**. Animal studies were conducted in accordance with the National University of Singapore (NUS) Institutional Animal Care and Use Committee (IACUC). The procedures used in this study were approved under the IACUC protocol. NOD SCID gamma (NSG) mice were obtained from the Center of Animal Resources (CARE) of NUS were bred under specific pathogen-free conditions. The mice were housed under controlled temperature (25 °C) and photoperiods (alternating 12 h of light and dark cycle) and were fed with standard diet. For each group of experiments, mice were age- and sex-matched.

Mice of 6–8 weeks of age were divided into 4–5 mice per group for use in the xenograft model experiment. HK-1 empty vector clone or DUSP16-overexpressing cells were harvested by trypsinization and resuspended in PBS. 0.1 mL of PBS containing $5 \times 10^6$ cells was inoculated on the left flank of each mouse. Cisplatin treatment started only when the tumor sizes were 0.125 cm$^3$. The mice were

injected intraperitoneally with either cisplatin (3 mg/kg) or an equal volume of PBS once every 3 days over 20 days. The weights and tumor surface areas of individual animals were recorded daily. Tumor surface area was calculated using the formula $W \times L$, where $L$ is the length and $W$ is the width of the tumor. At the end of the treatment, the animals were sacrificed by asphyxiation with $CO_2$. The tumors were excised and the weights of the tumors were recorded. The excised tumors were then fixed in 4% paraformaldehyde for histological analysis.

**Immunohistochemistry**. Immunohistochemistry was used for detection of cellular proteins or other antigens within cells and tissues using an antibody specific for the desired antigen. The fixed tumors were processed in a Leica TP1020 tissue processor (Leica Biosystems, Germany) as follows: 70% ethanol for 1 h, 80% ethanol for 1 h, 90% ethanol for 1 h, $3 \times 100\%$ ethanol for 1.5 h, $2 \times$ Histo-Clear (National Diagnostics, USA) for 2 h and $2 \times$ wax for 2.5 h. The processed tissues were subsequently embedded in paraffin, cut into 4.5-µm sections using a Leica RM2255 (Leica Biosystems), and mounted onto microscope glass slides. The mounted tissue sections were deparaffinized in Histochoice clearing agent (Sigma) for 15 min and rehydrated through graded alcohol washes (100% ethanol for 10 min, 95% ethanol for 10 min, and 70% ethanol for 10 min).

Antigen retrieval is subsequently carried out to break protein cross-links formed by the solvents, heat, and fixatives from fixation and paraffin embedding of tissues which reduces or prevents antibodies from binding to their antigens. This was carried out in sodium citrate buffer (10 mM sodium citrate, 0.05% Tween 20, pH 6.0) or EDTA (1 mM EDTA, 0.05% Tween 20, pH 8.0) for 20 min at boiling temperature.

The tissues are then treated with 3% hydrogen peroxide for 10 min to block endogenous peroxidase which reacts with diaminobenzidine (DAB) and causes background staining. Further blocking with 3% BSA in PBS for another 10 min was done to prevent the primary and secondary antibodies from binding non-specifically to the tissue section rather instead of the target antigen.

The individual tissue sections were stained overnight at 4 °C with primary antibodies against cleaved caspase 3 (Cell Signaling Technology) and biotin-conjugated Ki67 (eBioScience, USA) diluted 1:100 in 3% BSA in PBS. For cleaved caspase 3 detection, primary antibody incubation was followed by 2 h incubation with a biotinylated donkey anti-rabbit immunoglobulin G (H + L) (1:150 in 3% BSA in PBS) (ThermoScientific). This was followed by incubation with streptavidin-HRP secondary antibody (Biolegend, USA) (1:200 in 3% BSA in PBS) for 2 h. Peroxidase signal was developed using the chromagen, 3,3'-diaminobenzidine (DAB) (Dako, Denmark). Counterstaining was carried out by immersing the tissue sections in hematoxylin for 30 s, followed by submerging in water to obtain blue-stained nuclei to contrast with the brown positively stained antigens. Dehydration was carried out in 70% ethanol for 10 min, 95% ethanol for 10 min, 100% ethanol for 10 min, and Histochoice clearing agent for 15 min. After allowing the stained sections to dry overnight, the slides were mounted on coverslips using Histomount mounting reagent (Invitrogen). Total cleaved caspase 3 or Ki67 positive stained cells were expressed as percentage to the total number of nuclei per section.

**Clinicopathological and immunohistochemistry analyses**. Informed consent was obtained from patients before surgery for specimens to be used for research. Approval by the review board of National University of Singapore and the first affiliated hospital of Zhejiang Chinese Medical University was obtained prior to the commence of the study. Immunohistochemistry was performed on formalin-fixed, paraffin-embedded (FFPE) sections of HNSCC and breast tumor tissues. Immuno-scoring of clinical samples was performed by 1 pathologist and 1 scientist in a blinded manner based on the nuclear and/or cytoplasmic staining intensity (0, negative; 1, weak; 2, moderate; and 3, strong) and the proportion of stained tumor cells (0–100%). The final immunoreactive score (IRS) was determined by multiplying the intensity score with the fraction of stained cells (0–300). Clinical outcomes were defined by disease-free survival (DFS).

**Statistical analysis**. All data are presented as means ± standard error of the mean (SEM) unless otherwise indicated. Statistical analyses were done using two-tailed student's $t$-test in GraphPad Prism software (GraphPad Inc., CA) except for the analysis of association of DUSP16 with breast cancer patient survival which used IBM SPSS Statistic 19.0. $p < 0.05$ was considered statistically significant.

**Reporting summary**. Further information on research design is available in the Nature Research Reporting Summary linked to this article.

## Data availability

The authors declare that the data supporting the findings of this study are available within the article and its Supplementary Information files. All relevant data are available from the authors. Source data are provided with this paper.

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

## Acknowledgements

This work was supported by grants from the National Research Foundation, Prime Minister's Office, Singapore, under its Campus of Research Excellence and Technological Enterprise (CREATE) program (R571-010-012-592 to Y.Z.), the Singapore National Medical Research Council (NMRC/OFIRG/0059/2017 to Y.Z.), the NUS Global Asia Institute (R571-000-043-133 to Y.Z.), and the National Research Foundation (NRF-CRP19-2017-04 to N.R.J.G.). B.W. was supported by a scholarship from China Scholarship Council. We thank Dr. P. Hutchinson and Mr. G. Teo from the Flow Cytometry Lab of the Life Sciences Institute for assistance with flow cytometry.

## Author contributions

Y.Z., X.X., B.C.G., H.B.L., Z.L.W., and B.W. designed the research; H.B.L., Z.L.W., B.W., L.R.K., C.W.P., Y.C., C.Li, F.X., X.X., H.Y., J.M.L., F.Y.X.L., I.B.H.T., and R.D. performed research; H.B.L., Z.L.W., B.W., H.M.S., H.S., N.R.J.G., B.C.G., X.X., and Y.Z. analyzed data; H.B.L., Z.L.W., X.X., and Y.Z. wrote the paper.

## Competing interests

The authors declare no competing interests.
