## [Peer Review File · Nature Communications]

Reviewers' comments:

Reviewer #1 (Remarks to the Author); expert in platinum-drugs resistance:

The authors are congratulated on addressing a clinically relevant question pertinent to many cancers- platinum therapy resistance. through studies of cell lines from multiple sites of origin (nasopharyngeal, colorectal, gastric and others) they demonstrate that higher levels of DUSP16 is associated with platinum resistance and its knockout can sensitize tumor cells to platinum drugs. The hypothesis hinges on the idea that DUSP16 negatively regulates MAPK pathway. In vivo data using cell lines utilizing xenografts generated from cell lines suggests that there is marginal sensitization of platinum drugs when DUSP16 is knocked down and increase in platinum resistance when it is over expressed. The underlying mechanism is proposed to be through activation of apoptosis.

The following questions emerge

- 1) what was the rationale for studying DUSP16- can it be targeted therapeutically?
- 2) Is there any evidence that higher levels of this protein is associated with platinum resistance in clinical samples? figure 7D shows an IHC on a patient cohort with head and neck cancers but the numbers are small and are the treatment strategies matched/equivalent in the patient cohorts. Is there any data from TCGA that suggests DUSP-16 as relevant to platinum resistance?
- 3) Is it possible to evaluate the relevance of this protein using PDX models?
- 4) even if used solely as a biomarker how do the authors envision clinical translation of this work to stratify patients into those that have platinum resistant vs. platinum sensitive disease?

I appreciate the opportunity to review this manuscript.

Reviewer #2 (Remarks to the Author); expert in DUSP:

DUSP phosphatases as regulators of MAPK signaling are good candidates for a role in tumor cell growth and malignancy. Here, the authors investigate the expression and function of DUSP16, a p38/JNK-specific MKP, in various human cancer cell lines and find that high DUSP16 RNA and protein expression associates with reduced apoptosis upon platinum-based treatment. Overexpression of DUSP16 inhibits apoptosis, suggesting a causal effect, which was corroborated by CRISPR/Cas9 knockout of DUSP16 in C666-1 cells leading to enhanced JNK activation and apoptosis. Mechanistically, DUSP16 reduced p38/JNK activation, reduced Bax protein levels, mitochondrial potential loss, and cleavage of Caspases 3 and 9. In a NSG mouse tumor cell transfer model using DUSP16-expressing HK-1 cells, or DUSP16 KD in C666-1 cells, the effects on cisplatin-induced apoptosis was replicated in vivo. Importantly, in addition to the work with cancer lines, the authors provide evidence that DUSP16 expression levels in immunohistochemistry analysis of tissue samples from HNSCC patients correlate with reduced disease-free survival.

Overall, this work is of considerable interest as it identifies DUSP16 as a promoter of tumor cell resistance to chemotherapy that may be relevant for prognosis and could be targeted to enhance treatment efficacy by unleashing JNK activity promoting tumor cell death. The results are original in that to date several studies have suggested that DUSP16 acts rather as a tumor suppressing factor.

There are several points of criticism I have to make:

1. Novelty: In the introduction, the authors correctly state and reference that in the context of cancer, Dusp16 downregulation has been associated with different Tumor entities, but there is also previous evidence for a Tumor suppressive effect of Dusp16 silencing via increased JNK activation in Burkitt lymphoma (Lee Syed et al. 201 Br J Cancer). In fact, in that 2010 paper, evidence is provided that Dusp16 overexpression inhibits Cisplatin-induced JNK activation (Fig. 5B/C/E in Lee Syed 2010). This

prior published evidence of DUSP16-dependent attenuation of the response to Cisplatin needs to be cited more clearly in the manuscript.

2. Association of DUSP16 levels in primary human tumors with prognosis: the numbers of patient samples is not too high, and the p-value obtained in the survival analysis in Fig. 7D is just borderline significant. Including an analysis of Dusp16 mRNA levels in larger published data sets from tumor samples and their association with outcome should be tried and may help to provide further evidence.

3. Specific comments to the manuscript and figures:

a) Protein detection of DUSP16 by Western blot: as different isoforms for DUSP16 have been described, it is important to provide the molecular weight of the DUSP16 protein band. I assume that it corresponds to the DUSP16 A1 isoform of >70 kDa; however, images of the complete membranes should be shown in the Supplemental Data to clarify whether there are additional DUSP16-specific bands. This applies to Western blot data in Figs. 1, 2, and 6 (KD cell lines).

b) In Fig. 1E+F, only single data points are shown for DUSP16 protein and apoptosis rate. Provide information about reproducibility.

c) Line 238/9: the Statement "... that Dusp16 is an important mediator of sensitivity of cancer cells to cisplatin" is not correct here to summarize the results of overexpression in Fig. 2. Please delete.

d) Fig. 7A: the number of mice per group is given as "at least 3". This is a very small number to make statistical comparisons. In my opinion, the experiment also has to be repeated to get beyond preliminary character.

Reviewer #3 (Remarks to the Author); expert in CRC:

The manuscript from Low and colleagues describes a role for the dual-specificity phosphatase 16 (DUSP16) in mediating resistance from platinum-based chemotherapy. The authors show that DUSP16 over expression mediates cisplatin resistance, knockdown promotes drug sensitivity and DUSP16 expression modulates JNK and p38 activity. While this manuscript represents a significant body of work, I have numerous concerns that dampen my enthusiasm for this work.

1. Much if this work is descriptive. It is entirely unclear how DUSP16 modulates JNK or p38 activity or impinges upon mitochondrial function or cell death. Some mechanistic data describing how DUSP16 achieves these phenotypes is essential.

2. The central role for JNK and p38 in mediating the effects of DUSP16 are questionable, given the limited phenotypes seen following chemical modulation of p16 and JNK activity.

3. The tumor phenotypes driven by DUSP16 over expression and loss are quite minor. They are also not clearly apparent from tumor samples.

4. The specific connection to platinum therapy here is unclear. The authors should explore whether DUSP16 more broadly affects the response to cytotoxic stress.

5. The role of DUSP16 in modulating the cell cycle could account for the effects seen on cell death. This is not examined in this work.

Responses to Reviewers' Critiques

We thank the reviewers for their constructive comments and suggestions which helped us to improve our manuscript in a great way. In the past 6 months, we have performed numerous experiments to address the concerns from the reviewers. The experiments performed include analysis of a cohort of breast cancer patients treated with platinum drugs; xenograft experiment of vector- and DUSP16-expressing HK-1 cells using more immune deficient mice; cell cycle analysis of HK-1 cells in response to cisplatin treatment; examination of various types of cancer cells in response to other cytotoxic drugs including 5-fluorouracil (5-FU) and epirubicin; and *in vitro* dephosphorylation of activated JNK1/2 and p38 by DUSP16 protein. We have specifically addressed each of the reviewer's comments point by point as following:

Reviewer #1 (Remarks to the Author); expert in platinum-drugs resistance:

The authors are congratulated on addressing a clinically relevant question pertinent to many cancers- platinum therapy resistance. through studies of cell lines from multiple sites of origin (nasopharyngeal, colorectal, gastric and others) they demonstrate that higher levels of DUSP16 is associated with platinum resistance and its knockout can sensitize tumor cells to platinum drugs. The hypothesis hinges on the idea that DUSP16 negatively regulates MAPK pathway. In vivo data using cell lines utilizing xenografts generated from cell lines suggests that there is marginal sensitization of platinum drugs when DUSP16 is knocked down and increase in platinum resistance when it is over expressed. The underlying mechanism is proposed to be through activation of apoptosis.

The following questions emerge

1) what was the rationale for studying DUSP16- can it be targeted therapeutically?

Response: The answer to this question is "Yes, we believe that DUSP16 can be targeted therapeutically". Our study showed that higher DUSP16 is associated with increased resistance to platinum-based drugs. Therefore, DUSP16 inhibitors can be used to suppress DUSP16 expression in cancer cells to improve sensitivity to platinum-based drugs. We are planning to identify DUSP16-specific inhibitors through screening of commercially available compound libraries. In addition, small molecule inhibitors can be designed and synthesized based on the protein structure of DUSP16, which we will pursue in collaboration with colleagues in the Department of Chemistry in our school.

2) Is there any evidence that higher levels of this protein is associated with platinum resistance in clinical samples? figure 7D shows an IHC on a patient cohort with head and neck cancers but the numbers are small and are the treatment strategies matched/equivalent in the patient cohorts. Is there any data from TCGA that suggests DUSP-16 as relevant to platinum resistance?

Response: We thank the reviewer for the question and suggestion. We are delighted to let the reviewer know that we managed to obtain a cohort of 113 breast cancer patients who were treated with platinum-based chemotherapy. We examined the expression of DUSP16 protein in tumor samples from these patients by immunohistological staining and analysed the levels of DUSP16 expression with patient survival. The results showed that higher DUSP16 expression is significantly associated with reduced survival (Figure 1 below). The results were incorporated in the revised Figure 7E-F in the manuscript. For the suggestion on data from TCGA, we searched TCGA database and didn't find any platinum-based drug treated patient cohort.

Figure 1. Increased DUSP16 protein expression is associated with reduced survival in breast cancer patients treated with platinum drugs. (A). Representative images showing negative (0+), low (1+), intermediate (+2), and high (3+) DUSP16 protein expression determined by immunohistological staining in breast cancer tumour tissues. **(B).** Patient survival analysis in association with DUSP16 protein levels showing that higher DUSP16 expression was associated with significantly reduced survival ($P < 0.0001$).

3) Is it possible to evaluate the relevance of this protein using PDX models?

Response: Thanks for the suggestion. We managed to obtain two tumour samples from local hospital and used them to try to establish PDX model. Unfortunately, the tumour cells did not grow in immune deficient mice. One of the reasons for no growing of the tumour cells is perhaps the early stage of the tumour samples. We are still waiting for more patient tumour samples. Nevertheless, we believe that our breast cancer patient cohort analysis has provided important clinical relevance of this protein with platinum drug resistance.

4) even if used solely as a biomarker how do the authors envision clinical translation of this work to stratify patients into those that have platinum resistant vs. platinum sensitive disease?

Response: Thanks for the question. Our study demonstrates that patients with higher DUSP16 protein expression will have higher resistance to platinum drugs compared to patients with lower DUSP16 expression in their tumours. Therefore, the expression of DUSP16 in freshly resected tumour samples could be assessed using immunohistological staining. DUSP16 protein levels in tumour tissue samples in tissue bank can be used as reference to grade DUSP16 protein levels into negative, low, intermedia and high as we did in our study. The information obtained will help the clinicians in designing chemotherapy protocols for the patients. Our long-term goal is to develop therapeutic strategies targeting this molecule such as identification of DUSP16 specific inhibitors or using genetic approaches to reduce DUSP16 expression thereby re-sensitize patients who are resistant to platinum drugs due to DUSP16 protein expression.

I appreciate the opportunity to review this manuscript.

Response: Thanks a lot for your insightful questions and constructive suggestions.

Reviewer #2 (Remarks to the Author); expert in DUSP:

DUSP phosphatases as regulators of MAPK signaling are good candidates for a role in tumor cell growth and malignancy. Here, the authors investigate the expression and function of DUSP16, a p38/JNK-specific MKP, in various human cancer cell lines and find that high DUSP16 RNA and protein expression associates with reduced apoptosis upon platinum-based treatment. Overexpression of

DUSP16 inhibits apoptosis, suggesting a causal effect, which was corroborated by CRISPR/Cas9 knockout of DUSP16 in C666-1 cells leading to enhanced JNK activation and apoptosis. Mechanistically, DUSP16 reduced p38/JNK activation, reduced Bax protein levels, mitochondrial potential loss, and cleavage of Caspases 3 and 9. In a NSG mouse tumor cell transfer model using DUSP16-expressing HK-1 cells, or DUSP16 KD in C666-1 cells, the effects on cisplatin-induced apoptosis was replicated in vivo. Importantly, in addition to the work with cancer lines, the authors provide evidence that DUSP16 expression levels in immunohistochemistry analysis of tissue samples from HNSCC patients correlate with reduced disease-free survival.

Overall, this work is of considerable interest as it identifies DUSP16 as a promoter of tumor cell resistance to chemotherapy that may be relevant for prognosis and could be targeted to enhance treatment efficacy by unleashing JNK activity promoting tumor cell death. The results are original in that to date several studies have suggested that DUSP16 acts rather as a tumor suppressing factor.

Response: We thank the reviewer for the positive comments on the study.

There are several points of criticism I have to make:

1. Novelty: In the introduction, the authors correctly state and reference that in the context of cancer, Dusp16 downregulation has been associated with different Tumor entities, but there is also previous evidence for a Tumor suppressive effect of Dusp16 silencing via increased JNK activation in Burkitt lymphoma (Lee Syed et al. 2010 Br J Cancer). In fact, in that 2010 paper, evidence is provided that Dusp16 overexpression inhibits Cisplatin-induced JNK activation (Fig. 5B/C/E in Lee Syed 2010). This prior published evidence of DUSP16-dependent attenuation of the response to Cisplatin needs to be cited more clearly in the manuscript.

Response: We would like to let the reviewer know that this paper was actually cited in the manuscript (Reference 55). In the introduction (Page 7-8, line 125-128 in the first version of the manuscript), we stated that "There is also evidence that methylation-dependent transcriptional silencing of DUSP16 in Burkitt's lymphoma cells abrogated the negative regulation of JNK activity, and enhanced the cell sensitivity to chemotherapeutic agents that activate JNK, such as doxorubicin, sorbitol and cisplatin". We have highlighted this sentence in yellow in the marked revised manuscript for your peruse.

2. Association of DUSP16 levels in primary human tumors with prognosis: the numbers of patient samples is not too high, and the p-value obtained in the survival analysis in Fig. 7D is just borderline significant. Including an analysis of Dusp16 mRNA levels in larger published data sets from tumor samples and their association with outcome should be tried and may help to provide further evidence.

Response: We agree with the reviewer about the small size of head & neck cancer patient cohort. Thanks for the suggestion. To provide additional clinical relevance of DUSP16 with response to platinum drugs, we are lucky to obtain a cohort of breast cancer patients who were treated with platinum-based chemotherapy. Analysis of DUSP16 protein expression with patient survival capability was carried out. The results demonstrated that higher DUSP16 expression was significantly associated with poor patient survival ($P < 0.0001$) (Fig. 1 above and Figure 7 E-F in the revised Figures).

Following reviewer's suggestion on analysis of Dusp16 mRNA levels in larger published data sets from tumor samples, we found a cohort of low-grade gliomas (n=468) in which high expression of DUSP16 mRNA is significantly associated with reduced survival probability ($P=0.014$). In addition, in

two small cohorts of cancer patients, lymphoid neoplasm diffuse large B-cell lymphoma (n=48) and adenoid cystic carcinoma (n=76), high Dusp16 expression were also associated with reduced survival probability. However, the difference did not reach statistical significant, likely due to small number of patients in both cohorts. The results were presented in Figure 2 below.

Figure 2. DUSP16 mRNA expression with survival probability of three types of cancer. (A). Low-grade gliomas (n=468), **(B).** Lymphoid neoplasm diffuse large B-cell lymphoma (n=48), and **(C).** Adenoid cystic carcinoma (n=76) patient datasets from TCGA database were used to analyze the expression of DUSP16 mRNA in association with patient survival probability.

3. Specific comments to the manuscript and figures:

a) Protein detection of DUSP16 by Western blot: as different isoforms for DUSP16 have been described, it is important to provide the molecular weight of the DUSP16 protein band. I assume that it corresponds to the DUSP16 A1 isoform of >70 kDa; however, images of the complete membranes should be shown in the Supplemental Data to clarify whether there are additional DUSP16-specific bands. This applies to Western blot data in Figs. 1, 2, and 6 (KD cell lines).

Response: Thanks for the suggestion. The anti-DUSP16 anti-body (D5F4, rabbit mAb, Cat #5523) was purchased from Cell Signaling. This antibody only recognizes one DUSP16 band and the molecular weight is supposed to be 79 KDa (Figure 3A below). In our hand, we obtain a band at the size around 75 KDa. The small difference of the sizes of the bands is possibly due to the different protein ladders used and the gel running. We have attached a full gel western picture in Figure 3B and a half gel picture in Figure 3C below. In Figure 3B, we used HCT116 cells (express minimal level of DUSP16) with vector and DUSP16 overexpression to perform the western blot. In Figure 3C, we used HK-1 cells with vector and DUSP16 overexpression to perform the western. The DUSP16 western blot results were consistent in all the experiments performed using this antibody. Due to the amount of data and the limitation of space, we will not show these images in the figures. We thank the reviewer for understanding.

Figure 3. Anti-DUSP16 antibody recognizes a band close to 79kDa. (A). Reference image of anti-DUSP16 western blot from Cell Signaling. **(B).** Vector- and DUSP16-overexpression HCT116 cell clones were used to detect DUSP16 protein expression. **(C).** DUSP16 protein expression in vector- and DUSP16-overexpression HK-1 cells was examined by western blot.

b) In Fig. 1E+F, only single data points are shown for DUSP16 protein and apoptosis rate. Provide information about reproducibility.

Response: We have included a set of data of cell apoptosis in response to cisplatin at 0, 24 and 48 hours for CRC cell line DLD1 and HCT116 as well as gastric cancer cell lines Ags and Nugc3 (Figure 4 below). The results were included in Supplementary Figure 1C-D.

Figure 4. High DUSP16 protein expression is associated with increased resistance to cisplatin. (A). Colorectal cancer cell line DLD1 (DUSP16 high) and HCT116 (DUSP16 low), and **(B),** gastric cancer cell line Ags (DUSP16 high) and Nugc3 (DUSP16 low) were treated with cisplatin for 0, 24 and 48 hours to examine cell apoptosis by Annexin V and 7AAD staining followed by flow cytometry analysis.

c) Line 238/9: the Statement "... that Dusp16 is an important mediator of sensitivity of cancer cells to cisplatin" is not correct here to summarize the results of overexpression in Fig. 2. Please delete.

Response: Following reviewer's suggestion, we have deleted this sentence from the manuscript.

d) Fig. 7A: the number of mice per group is given as "at least 3". This is a very small number to make statistical comparisons. In my opinion, the experiment also has to be repeated to get beyond preliminary character.

Response: Thanks for pointing out this. This experiment has been repeated for three times with at least 3 mice in each group in each experiment and similar results were obtained. For the first experiment, 3 mice in the two vector groups (no treatment and cisplatin treatment) and 3 mice in DUSP18 knockdown (KD) no treatment group and 4 mice in KD group with cisplatin treatment were used. In the 2nd and 3rd experiments, 3 mice were used in each group. Therefore, in total, we have used 9 mice in each of the vector groups, 9 and 10 mice in DUSP16 KD no treatment group and cisplatin treatment group respectively.

Reviewer #3 (Remarks to the Author); expert in CRC:

The manuscript from Low and colleagues describes a role for the dual-specificity phosphatase 16 (DUSP16) in mediating resistance from platinum-based chemotherapy. The authors show that DUSP16 over expression mediates cisplatin resistance, knockdown promotes drug sensitivity and DUSP16 expression modulates JNK and p38 activity. While this manuscript represents a significant body of work, I have numerous concerns that dampen my enthusiasm for this work.

1. Much if this work is descriptive. It is entirely unclear how DUSP16 modulates JNK or p38 activity or impinges upon mitochondrial function or cell death. Some mechanistic data describing how DUSP16 achieves these phenotypes is essential.

Response: We thank the reviewer for the suggestion. To address this comment, we have performed in vitro dephosphorylation assay in which phosphorylated JNK1/2 or p38 protein was incubated with

recombinant DUSP16 protein followed by western blot analysis to assess the level of phosphorylation of JNK1, JNK2 and p38. As shown in Figure 5 below, DUSP16 was able to dephosphorylate JNK1, JNK2 or p38. The data was also incorporated Supplementary Fig. 5B in the revised manuscript. To further substantiate the link between DUSP16-JNK/p38 and Bax, we planned to perform in vitro phosphorylation assay of Bax by pJNK1/2 or pP38. Unfortunately, phosphor-Bax antibody which can be used to detect the phosphorylation site in Bax by JNK/p38 using western blot is not available commercially. We purchased a phosphor-Bax antibody and tried several rounds of optimization of the western blot condition in the hope of detecting phosphor-Bax but failed. With the lockdown of the lab due to the COVID-19 situation, we were unable to continue to try the western blot. However, we believe that the results of the dephosphorylation assay provide further evidence on DUSP16 in regulation JNK/p38-Bax signalling, thereby regulating cell death.

Figure 5. DUSP16 is able to dephosphorylate JNK1/2 and p38 in vitro. HEK293 T cells were transfected with Flag-tagged JNK1, JNK2 or p38 (3 μ g DNA each). Cells were stimulated with 100 ng/ml EGF for 10 min to induce MAPKs phosphorylation before total protein was harvested. The Flag-tagged proteins were purified by immunoprecipitation using anti-Flag magnetic beads (Sigma) and used in the respective dephosphorylation reactions by incubation with recombinant DUSP16 protein in dephosphorylation buffer at 37 $^{\circ}$ C for 2.5 hours and phosphorylation was analyzed by western blot.

2. The central role for JNK and p38 in mediating the effects of DUSP16 are questionable, given the limited phenotypes seen following chemical modulation of p16 and JNK activity.

Response: We have examined the activation of STAT3 and AKT, the expression of cyclin D1 and p53 in vector- and DUSP16-expression NPC cell HK1 and CRC cell HCT116 in response to cisplatin at various time points. The results below showed there is no difference in the activation/expression of these molecules between vector- and DUSP16-expressing cells (Figure 6A). In addition, we performed proteomic analysis of vector- and DUSP16-expression DLD1 cells at 6 hours upon cisplatin treatment. Analysis of differentially expressed proteins demonstrated that MAPK pathway is the one with the highest number of proteins differentially expressed between vector- and DUSP16-expressing cells. In addition, we performed JNK1/2 and p38 dephosphorylation assay by DUSP16 and the results demonstrated that DUSP16 was able to dephosphorylate JNK1, JNK2 and p38 (Figure 5 above). We believe that the phenotypes we reported is mainly due to the regulation of JNK and p38 by DUSP16. We will further examine the contribution of other possible molecules regulated by DUSP16 in drug resistance in cancer in future.

Figure 6. (A). Vector- and DUSP16-overexpressing HK1 and HCT116 cells were stimulated with cisplatin for the indicated duration and cells were harvested for western blot analysis of activation of STAT3, AKT and expression of cyclin D1 and p53. (B). Vector- and DUSP16-overexpressing DLD1 cells were treated with cisplatin (30 μ g/mL) for 6 hour. Cell lysates were digested and labelled with iTRAQ followed by mass spectrometry analysis. Differentially expressed proteins were analysed.

3. The tumor phenotypes driven by DUSP16 over expression and loss are quite minor. They are also not clearly apparent from tumor samples.

Response: Each xenograft experiment was repeated for three times and in each experiment the difference of the tumour size between vector- and DUSP16-overexpressing cells was significance. In addition, we have repeat one experiment, the vector- and DUSP16-overexpressing HK1 cell xenograft with 6 mice in each experimental group. The results confirmed that overexpression of DUSP16 resulted in resistance of the tumour cells to cisplatin (Figure 7 below and revised Figure 5A in the manuscript). Furthermore, our analysis of breast cancer patients with platinum drug treatment demonstrated that high DUSP16 was significantly associated with reduced survival probability (Figure 1 above and revised Figure 5A in the manuscript).

Figure 7. Overexpression of DUSP16 in HK-1 cells results in resistance to cisplatin in vivo. Vector- or DUSP16-transfected HK-1 cells were inoculated into NSG mice. Cisplatin was injected at 3mg/kg body weight every 3 days. After 20 days, tumours were harvested to measure the sizes and weights. The data are expressed as dots representing the weight of each tumour; the scale bars are the mean \pm standard error of the mean (SEM) of tumor weights from 6 mice in each treatment group of the representative experiment. ** $P < 0.01$ (mean \pm SEM, $n = 6$).

4. The specific connection to platinum therapy here is unclear. The authors should explore whether DUSP16 more broadly affects the response to cytotoxic stress.

Response: Thank for the suggestion. Following this suggestion, we treated vector- or DUSP16-overexpressing CRC cell line HCT116, NPC cell line HK-1, gastric cancer cell line NUGC3 and breast cancer cell line MDA-MB231 with 5-FU or epirubicin to examine apoptosis. As shown in Figure 8A

below, overexpression of DUSP16 resulted in resistance of the cell to 5-FU and epirubicin. In addition, decreased activation of JNK and p38, decreased expression of cleaved caspase 9 and caspase 3 were observed in HCT116 and NUGC3 cells with DUSP16 overexpression compared to respective vector transfected control cells (Figure 8B). These results demonstrated that DUSP16 indeed more broadly affect the response to cytotoxic stress, which warrants further investigation. We have presented the data in Supplemental Figure 7 in the revised manuscript.

Figure 8. Overexpression of DUSP16 resulted in resistance to 5-FU and epirubicin in various cancer cells. (A-B). Vector- and DUSP16-expressing HCT116, HK-1, NUGC3 and MDA-MB231 cells were treated with 5-FU (A) or epirubicin (B) for 48 hours to determine cell apoptosis. **(C-D).** Activation of JNK, p38 and ERK, and expression of cleaved caspase 9 (C-C9) and caspase 3 (C-C3) in HCT116 cells (C) and NUGC3 (D) was analysed by western blot.

5. The role of DUSP16 in modulating the cell cycle could account for the effects seen on cell death. This is not examined in this work.

Response: Thanks for the suggestion. We have carried out cell cycle analysis of vector- and DUSP16-expressing HK-1 cells in response to cisplatin treatment at various time points by propidium iodide (PI) staining and flow cytometry analysis. The results showed (Figure 9) comparable percentage of G1 cells between vector- and DUSP16-expressing HK-1 cells, suggesting that DUSP16 is dispensable in regulation of cell cycle in response to cisplatin.

Cisplatin (5ug/mL)

Figure 9. Cell cycle analysis of HK-1 cells with or without DUSP16 overexpression in response to cisplatin. Vector- or DUSP16-expressing HK-1 cells were treated with cisplatin for the indicated period of time. Cell cycle analysis was carried out by propidium iodide staining followed by flow cytometry.

Reviewers' comments:

Reviewer #1 (Remarks to the Author):

Thank you for addressing my questions.

Reviewer #2 (Remarks to the Author):

The authors have addressed my questions in Detail and convincingly. The inclusion of the breast cancer data confirmed the association of high Dusp16 Expression with low survival.

Reviewer #3 (Remarks to the Author):

In the revised manuscript, the authors have addressed some issues raised in the initial review. The most compelling new data is that DUSP16 levels correlate with chemotherapeutic outcome in breast cancer (Figure 7F). Based on this data, DUSP16 may indeed serve as a biomarker for outcome in this disease. The expression data in glioma is not convincing.

I still, however, have some major concerns that remain from the initial review.

First, the phenotypes are quite minor. The tumor data in 5B and 7A do not suggest a striking difference following modulation of DUSP16 levels. The data may still result from the ability of DUSP16 to regulate the cell cycle. Indeed, the revised data show an increase in G1 cells (and, apparently, G2 cells) upon DUSP16 over expression. While these changes are minor, so are the apoptosis phenotypes reported (6C, for example).

Second, if the argument is that DUSP16 directly dephosphorylates p38 and JNK, this has to be shown using more decisively (or should be omitted from the manuscript).

Finally, the manuscript repeatedly argues that DUSP16 modulates the response to cisplatin. Yet, the experiments performed during the paper revision shows that the effect is seen in response to numerous chemotherapies. Indeed, there is nothing in the manuscript to suggest a specific mechanistic link to cisplatin.

Responses to Reviewers' Critiques

We are delighted that both Reviewer 1 and Reviewer 2 were satisfied with our revised manuscript and are very happy to know that Reviewer 3 is also convinced that DUSP16 indeed can serve as a biomarker for outcome of cancer especially breast cancer. In the past 5 months, we have performed additional experiments including examination of interaction between JNK, p38 and Bax by immunoprecipitation, generation of Bax knockout DLD1 cells using CRISP/cas9 technology and using Bax inhibitor Peptide V5 to treat various types of cancer cells upon cisplatin treatment to further test the role of Bax in DUSP16-mediated cisplatin resistance. We now address Reviewer 3's comments point-by-point below.

Reviewer #3 (Remarks to the Author):

In the revised manuscript, the authors have addressed some issues raised in the initial review. The most compelling new data is that DUSP16 levels correlate with chemotherapeutic outcome in breast cancer (Figure 7F). Based on this data, DUSP16 may indeed serve as a biomarker for outcome in this disease. The expression data in glioma is not convincing.

I still, however, have some major concerns that remain from the initial review.

First, the phenotypes are quite minor. The tumor data in 5B and 7A do not suggest a striking difference following modulation of DUSP16 levels. The data may still result from the ability of DUSP16 to regulate the cell cycle. Indeed, the revised data show an increase in G1 cells (and, apparently, G2 cells) upon DUSP16 over expression. While these changes are minor, so are the apoptosis phenotypes reported (6C, for example).

Response: I respectfully disagree with this assessment. In Fig 5B, we showed that in DLD-1 CRC cells, the average tumor size of DUSP16 overexpression cells was 470mg (DUSP16-treated), whereas that of vector-transfected cells was 280mg (vector-treated) in cisplatin-treated cells (Fig 5B), meaning that the average weight of vector-transfected tumors was 67.9% heavier than that of DUSP16-expressing tumors. In Fig 7A, the results showed that knockdown of DUSP16 in C666-1 cell resulted in reduction of average tumour size from 576mg (vector) to 323mg (knockdown) in response to cisplatin treatment (43.9% reduction).

For the cell cycle analysis, we analyzed CRC HCT116 cells in response to cisplatin treatment. The results again showed comparable percentage of G1 cells (Figure 1 below). We therefore have to conclude that alteration of cell cycle is unlikely plays an important role in DUSP16-mediated cancer cell resistance to cisplatin and other platinum-drugs.

Figure 1. Cell cycle analysis of HCT116 cells with or without DUSP16 overexpression in response to cisplatin. Vector- or DUSP16-expressing HK-1 cells were treated with cisplatin for the indicated period of time. Cell cycle analysis was carried out by propidium iodide staining followed by flow cytometry.

Reviewer #3: *Second, if the argument is that DUSP16 directly dephosphorylates p38 and*

JNK, this has to be shown using more decisively (or should be omitted from the manuscript).

Response: DUSP16, also known as MKP7, has been known for 20 years to interact with and inactivate JNK and p38¹. We also tested the interaction between DUSP16 and JNK/p38. As shown in Figure 2 below, DUSP16 expression in HEH293 T cells is able to pull-down endogenous JNK and p38 (Figure 2 below and the data was incorporated in Figure S5A in the revised manuscript), but not Bax, demonstrating the interaction between DUSP16 and JNK/p38. Together with the result showing that DUSP16 is able to directly dephosphorylate both JNK1/2 and p38 (Fig S5B in the revised manuscript), we believe that we have provided convincing evidence demonstrating the direct regulation of JNK/p38 by DUSP16.

Our study shows that DUSP16 regulates Bax through JNK/p38 in multiple types of cancer cells, thereby mediating cancer cell resistance to cisplatin and other platinum-drugs. To further substantiate the role of Bax in DUSP16-mediated resistance to platinum-based chemotherapy, after discussion with our editor, Dr Garcia-Fernandez, in early June, we decided to take two approaches to address if DUSP16 truly regulates apoptosis through Bax. The first was to use Bax inhibitor to treat Vector- and DUSP16-overexpressing cells in response to cisplatin. The other was to knockout Bax in vector- and DUSP16-overexpressing DLD-1 cells to test their response to cisplatin. Now we have completed both experiments. As shown in Figure 3A below (has been included into Fig. 4G in the revised manuscript), Bax was successfully knocked out in both Vector- and DUSP16-overexpressing DLD1 cells (Fig. 3A). Treatment with 30 μ g/mL of cisplatin resulted in death of more than 50% (Annexin-V⁺ plus Annexin-V⁺7AAD⁺) of vector-expressing cells (Fig. 3B below and Fig. 4H in the revised manuscript), whereas cell death of DUSP16-overexpressing cells was below 30%. Knockout (KO) of Bax in vector-expressing cells resulted in a decrease of cell death to 26.3%. In contrast, the percentage of cell death of DUSP16-overexpressing/Bax-KO cells remained similar to that of DUSP16-overexpressing cells (27.5% vs 28.8%), comparable to that in vector-expressing cells after Bax was knocked out. These results demonstrated that DUSP16 indeed regulates apoptosis through Bax. Consistently, vector-overexpressing/Bax-KO, DUSP16-overexpressing and DUSP16-overexpressing/Bax-KO cells have comparable percentages of live cells (68.5%, 69.0% and 69.2%) after cisplatin treatment. These results demonstrate clearly that Bax plays an essential role in DUSP16-mediated cisplatin resistance in colorectal DLD-1 cells. Studies using Bax inhibitor Peptide V5 in various types of vector- and DUSP16-overexpressing cancer cells including NPC HK-1, CRC DLD-1, gastric cancer NUGC3 and breast cancer MDA-MB231 cells further supported the essential role of Bax in DUSP16-mediated cisplatin resistance (Fig. 4 below and Fig. S4 in the revised manuscript). Together, these results demonstrated that DUSP16 regulates JNK/p38-Bax signalling to inhibit apoptosis in various types of cancer cells (Fig. 5 below and Fig. S7 in the revised manuscript).

Figure 2. DUSP16 interacts with JNK and p38, but not Bax. 293T cells were transfected with Flag-DUSP16 constructs or Flag-vector. 24 hours after transfection, cells were harvested for immunoprecipitation followed by western blot to examine its interaction with JNK, p38 and Bax.

Figure 3. Bax is essential for DUSP16-mediated cisplatin resistance in DLD-1 cells. **A.** Generation of bax knockout (KO) cells from vector- or DUSP16-overexpression cells. To generate Bax-KO cells, the following pair of short guide-RNA (sgRNAs) were ordered from Integrated DNA Technologies: Forward: 5'– CACCGGTTTCATCCAGGATCGAGCA –3'; Reverse: 5'– AAAGTCTCGATCCTGGATGAAACC –3'. Oligonucleotides for guide RNAs were cloned into the pSpCas9(BB)-2A-GFP (PX458) plasmid (Addgene plasmid #48138) as described by Ran et al. ² BAX sgRNA plasmids were then transfected into DLD-1 cells using Lipofectamine®3000. 24-48 hours after transfection, GFP positive cells were sorted and seeded individually into 96-well plates. BAX-knockout in clones selected were confirmed by western blot. **B.** Vector- and DUSP16-overexpression DLD1 cells with or without Bax KO were treated with cisplatin (30 μg/mL) for 48 hours followed by analysis of cell apoptosis by Annexin-V and 7-AAD staining and flow cytometry. Percentage of Annexin-V⁺ and Annexin-V⁺7-AAD⁺ cells were quantified. The data is representative of 2 experiments with similar results. *** $P \leq 0.001$.

Figure 4. Bax is essential for DUSP16-mediated cisplatin resistance in various types of cancer cells. **A.** HK-1 cells were treated with cisplatin with or without Bax inhibitor Peptide V5 (300mM). Cells were harvested at the indicated time points post-treatment to isolate mitochondria for examination of Bax accumulation in the mitochondria by western blot. Vector- and DUSP16-expressing NPC HK-1 cells (**B**), CRC DLD-1 cells (**C**), gastric cancer NUGC3 cells, and breast cancer MDA-MB231 cells (**E**) were treated with cisplatin with or without Bax inhibitor (Baxi) for 48 hours for examination of cell death by Annexin-V and 7-AAD staining followed by flow cytometry analysis. The data is representative of 2-3 experiments with similar results. *** $P \leq 0.001$.

Figure 5. DUSP16 regulates p38/JNK-Bax signalling to mediate platinum-drug resistance in cancer. Platinum-drug treatment of cancer cells results in activation of JNK and p38 to promote Bax activation and translocation to mitochondria for apoptosis. The treatment also induces the expression of DUSP16 which dephosphorylates both JNK and p38 to negatively regulate apoptosis. High expression of DUSP16 could therefore lead to drug resistance.

Reviewer #3: Finally, the manuscript repeatedly argues that DUSP16 modulates the response to cisplatin. Yet, the experiments performed during the paper revision shows that the effect is seen in response to numerous chemotherapies. Indeed, there is nothing in the manuscript to suggest a specific mechanistic link to cisplatin.

Response: In our first revision, following reviewer 3’s suggestion to “explore whether DUSP16 more broadly affects the response to cytotoxic stress”, we treated various vector- and DUSP16-overexpressing cancer cells with 5-FU and epirubicin. It was found that overexpression of DUSP16 resulted in resistance of cancer cells to these cytotoxic drugs too (Supplementary Fig. 6), demonstrating that DUSP16 indeed more broadly affects the response to cytotoxic stress and warrants further investigation. However, we don’t see why “non-specific mechanistic link to cisplatin” is an issue. What is important is the potential of this novel mechanism underlying cancer drug-resistance in clinical translation, which can improve efficacy of the treatment and patient outcomes.

Cisplatin and other platinum drugs remain the first-line agents against various types of cancers even in the era of precision medicine and immunotherapy³. Understanding the mechanism of resistance to platinum drugs is a major field of investigation. Effective treatment of metastatic cancers usually requires the use of toxic chemotherapy, and in most cases, multiple cytotoxic drugs. Elucidation of mechanisms that confer simultaneous resistance to different drugs has been a major goal in cancer biology, with the hope to overcome this major hurdle in cancer treatment to achieve durable disease control in patients with advanced cancers. Therefore, it is plausible that DUSP16 is not only responsible for platinum drug resistance but also important for resistance to other chemotherapy drugs, which requires future validation.

References

1. Tanoue, T., Yamamoto, T., Maeda, R. & Nishida, E. A Novel MAPK phosphatase MKP-7 acts preferentially on JNK/SAPK and p38 alpha and beta MAPKs. *J Biol Chem* **276**, 26629-26639 (2001).
2. Ran, F.A. *et al.* Genome engineering using the CRISPR-Cas9 system. *Nat Protoc* **8**, 2281-2308 (2013).

3. Rottenberg, S., Disler, C. & Perego, P. The rediscovery of platinum-based cancer therapy. *Nat Rev Cancer* (2020).

Reviewer's Comments:

Reviewer #3 (Remarks to the Author):

The revised manuscript includes considerable additional data - and has been improved by these additions.

Given the lengthy review process for this study, I certainly do not want to impede publication of this significant body of work.

For editorial purposes, I still am concerned about the extensive and specific referral to platinum drugs in this work. Cisplatin and platinum drugs are mentioned dozens of times in this manuscript - including a discussion of platinum mechanism and mechanisms of platinum resistance. Yet, the authors have evidence that the mechanism of DUSP16 chem-protection is not cisplatin specific. For the community reading this work, it absolutely matters whether DUSP16 should be evaluated as a general mechanism of chemo resistance or should be specifically evaluated in the context of cisplatin.

Response to reviewer's comments

REVIEWERS' COMMENTS

Reviewer #3 (Remarks to the Author):

The revised manuscript includes considerable additional data - and has been improved by these additions.

Given the lengthy review process for this study, I certainly do not want to impede publication of this significant body of work.

For editorial purposes, I still am concerned about the extensive and specific referral to platinum drugs in this work. Cisplatin and platinum drugs are mentioned dozens of times in this manuscript - including a discussion of platinum mechanism and mechanisms of platinum resistance. Yet, the authors have evidence that the mechanism of DUSP16 chem-protection is not cisplatin specific. For the community reading this work, it absolutely matters whether DUSP16 should be evaluated as a general mechanism of chemo resistance or should be specifically evaluated in the context of cisplatin.

Response: We thank the reviewer for the comments. We have revised the title of our manuscript to "DUSP16 promotes cancer chemoresistance through regulation of mitochondria-mediated cell death" and changed "platinum-drug" to "chemotherapeutic agents" or "chemotherapy drugs" to reflect the fact that DUSP16 regulates chemoresistance more broadly rather than cisplatin specific.